# Environmental Stress Testing for China's Overseas Coal Power Investment Project

**Minpeng Xiong [1], Xiaowen Yang [1], Sisi Chen [2,\*], Fulian Shi [1] and Jiahai Yuan [1,3,\*]**

[1] School of Economics and Management, North China Electric Power University, Beijing 102206, China; 50600182@ncepu.edu.cn (M.X.); 1182206066@ncepu.edu.cn (X.Y.); sfl@ncepu.edu.cn (F.S.)

[2] School of Education, University of Michigan, Ann Arbor, MI 48109, USA

[3] Beijing Key Laboratory of New Energy and Low-Carbon Development, North China Electric Power University, Beijing 102206, China

\* Correspondence: sschen1212@gmail.com (S.C.); yuanjiahai@ncepu.edu.cn (J.Y.)

**Abstract:** The advance of the Chinese "Belt and Road" initiative encourages increased overseas investment in coal power projects. However, it also brings about external environmental risks. In this paper, we use the approach of environmental stress testing to examine China's overseas coal power investment projects by focusing on two countries: Indonesia and Vietnam. We first identify five key testing factors (i.e., coal price, utilization hours, exchange rate, carbon tax, and environmental protection requirements) by examining the market regulation and the environmental risks of coal power projects along the "Belt and Road" countries. Then, we observed changes in the enterprise value and internal rate of return (IRR) by setting different scenarios in which the values of the five stress factors varied. The results show that (1) the economics of coal-fired projects in Indonesia is most sensitive to exchange rate, while the economics of coal projects in Vietnam is most sensitive to coal price; (2) the pressure of nationally determined contributions (NDC) goals on environmental protection will push the "Belt and Road" countries to implement more stringent environmental regulation, which will reinforce environmental stress on overseas coal power investment. These results have important policy implications for the enterprise, industry, and Chinese government.

**Keywords:** "Belt and Road" initiative; overseas investment; coal power project; environmental stress testing

## 1. Introduction

In 2013, China proposed the "Belt and Road" initiative (BRI) in order to promote common prosperity and development through enhanced inter-country trust and cooperation [1]. By the end of 2018, China had signed 170 intergovernmental cooperation documents with 122 countries and 29 international organizations, covering Asia, Africa, Europe, Oceania, and Latin America [2]. The investment and cooperation relationships that China have built with "Belt and Road" countries have driven the rapid economic development of many related countries. Data show that in the past five years, China's trade with "Belt and Road" countries generated more than 5 trillion U.S. dollars, with an average annual growth of 1.1%. The direct investment in these countries has reached more than 70 billion U.S. dollars, with an average annual growth of 7.2% [3]. With investments across all sectors, the energy sector is recognized as the basis for regional development. On 12 May 2017, the National Development and Reform Commission (NDRC) and the National Energy Administration (NEA) jointly released the "Vision and Action for Promoting Energy Cooperation between the Silk Road Economic Belt and the 21st Century Maritime Silk Road". This policy helps improve the global energy governance structure and promotes the goal of sustainable energy for all people by advocating

inter-country policy communication, unimpeded trade, and cooperation in energy investment, capacity, and infrastructure [4].The energy utilization technology and power industry levels in the "Belt and Road" countries generally lag behind other countries, and their electricity demand is increasing. In 2015, these countries covered a total population of 4.6 billion, but the per capita electricity consumption was about 2825 kWh, which was far below the international level of 3295 kWh [5]. In recent years, coal power has been developed in "Belt and Road" countries. In 2015, coal-fired power installations in these countries reached 1398 GW, accounting for 73% of the total coal-fired installed capacity worldwide. With all of the limitations of coal power in China, the continuing development of coal power in these countries opens up business opportunities for Chinese power companies. By the end of 2016, Chinese power companies including the State Grid, Huaneng Group, and Guodian Corporation had invested in 240 coal-fired power projects with a total installed capacity of 25.1 TW.

However, in recent years, as the issue of environmental protection has attracted wide international attention, Chinese power enterprises are facing great controversy when investing in overseas coal power projects. In 2015, nearly 200 Parties unanimously reached the "Paris Agreement" at the Paris Climate Change Conference, setting strict temperature-control standards after 2020 in response to global warming [6]. As one of the signatories to the agreement, China is suspected of having "export carbon emissions" for investing in coal power overseas. A new report by the Institute for Energy Economics and Financial Analysis (IEEFA) shows that China plans to build more coal-fired power plants overseas than Germany already has, arguing that the move goes against the global call to decarbonize [7]. In fact, energy infrastructure is an important guarantee for achieving the Sustainable Development Goals (SDGs), while the level of energy infrastructure in the "Belt and Road" countries is generally lagging behind. For many "Belt and Road" countries, considering the resource constraints, rapid growth of power demand, and price factors, their power supply structure in the future will still be dominated by coal power. First, China can help ease the pressure on their electricity demand. Second, most of the coal power plants China has invested in overseas have adopted advanced technologies and met local pollution discharge standards [8]. In addition, since 2016, China has significantly reduced its overseas coal power investment, and shifted more to clean coal power generation and renewable energy.

From the Chinese side, the power sector's enthusiasm in overseas coal power investment is largely driven by economic concern: coal power projects are a capital intensive investment and its return is stable and substantial. This is especially attractive for Chinese coal power utilities who are accustomed with the stable return expectations within a highly planned power system regulation under strong market growth [9]. However, with the recent advent of a new economical normal, power demand growth has slowed down and renewable energy is growing quickly in China, which has led to overcapacity in coal power and a radical structural change in the power market [10]. As a result, power utilities have switched to overseas investment. However, from an international perspective, China's active involvement in BRI coal power investment represents a big challenge to global efforts to stabilize greenhouse gas emissions, given the fact that industrialized economies have reached a consensus on divestment in coal. Though Chinese economic consideration and the climate change concern of the international society are not easily reconciled, a deep dive into Chinese economic consideration can shed more insight into the emissions abatement vision. As a matter of fact, what has happened to coal (divestment) in industrialized economies and what the impact of structural change to coal power will be in China will certainly happen in BRI developing economies sometime later. Therefore, even from a Chinese perspective, a more considerate strategy incorporating the potential risks of structural and market regulation changes in these BRI host countries and the long-term environmental and climate change risks in the decision-making process can possibly reconcile the pervasive contradiction. Therefore, it is necessary to foresee future environmental risks for these coal power projects which have been put into operation or will be put into operation. These include not only the risks related to the natural environment, but also those related to the "Belt and Road" national political environment and financial environment. In terms of the natural environment, the operation of coal power will be affected



by the abundance and quality of coal resources. For the political environment, in order to achieve the proposed NDC goals, each country will inevitably raise environmental protection standards, increase carbon and environmental protection taxes, and consider revisions to coal power planning. The risk to the financial environment mainly comes from the exchange rate fluctuations of "Belt and Road" countries. Taking Indonesia and Vietnam as examples, Indonesia's coal resources are rich but of low quality, while Vietnam's coal is more dependent on exports; Indonesia's exchange rate is volatile, while Vietnam's is more stable. It is important to incorporate these risk factors into China's investment decision-making in overseas coal power.

Stress testing is an environmental risk-analysis tool used to identify, define, and quantify risks. It was initially used in the financial sector to guide financial institutions in making investment decisions in high environmental-risk projects. The approach was to measure the impact of risk factors (e.g., climate change, air pollution, and government policy changes) on financial return [11]. Later, it was used to assess environmental risks in other industries. This paper applied the environmental stress test to China's overseas investment in coal power projects, and analyzed the influence of environmental risk on the economy of projects in Indonesia and Vietnam. These were chosen as the case nations because they are the largest coal power FDI investment destination countries of China. Though our purpose is not for comparative study, the difference in environmental factors of these two countries can reveal the sensitivity of stress testing due to differences in national status, market, and regulations.

The rest of this paper is organized as follows: Section 2 introduces the existing literature related to overseas power investment risk, the role of financial institutions, and stress test method. Section 3 identifies the environmental stress test factors by conducting an environmental risk analysis and constructing the corresponding environmental stress test conduction map. Section 4 quantifies the variation of these factors' influence on Indonesia and Vietnam, respectively, by conducting sensitivity tests of the enterprise value to environmental factors. Section 5 presents the empirical results. Section 6 provides our conclusions and policy recommendations at the national, industry, and enterprise levels in order to promote "Belt and Road" power cooperation.

## 2. Literature Review and Research Background

In this section, we point out the importance of studying the environmental risk of overseas coal power projects and quantifying the environmental risk from the participation background of financial institutions. Additionally, we present the feasibility of this method based on the available stress test research.

(1) Overseas Power Investment Risk

Many scholars have studied the risk of China's overseas power investment at the macro level. For example, Ling analyzed the political, economic, and security risks of China's investment in power projects in Pakistan [12]. Dong pointed out that the gaps between the countries along the "Belt and Road" in political, cultural, economic, and social development would cause financial risks to China's overseas investment [13]. Liu and Yan analyzed the influencing factors of power investment risk and introduced a fuzzy comprehensive evaluation model [14]. Farfan et al. proposed national sustainable development indicators of the power industry to measure the investment risks of the power industry in various countries [15].

In addition, some scholars have analyzed China's coal power investment risk. Yuan J. et al. [10,16] and Zhao et al. [9,17] analyzed the economics of coal power under a changing market landscape. Guo and Wang set up a risk decision model of coal power investment based on the risk preference of investors. They determined the objective function of the coal power investment risk decision based on the portfolio theory and expected utility function theory, and analyzed the expected return and risk of the coal power project investment scheme by the Monte Carlo method. Yuan and Li et al. evaluated the overseas investment risk of coal-fired power plants in countries along the Belt and Road initiative. The results showed that Singapore had the lowest risk of coal power investment, followed

by New Zealand and Thailand [18]. Kun and Fang discussed the advantages of China's thermal power technology and the drawbacks of overseas investment in environmental protection [19]. Yuan proposed that the "Belt and Road" power cooperation should adhere to the principle of "green and sustainable", help countries along the line to improve energy efficiency, and strive for the coordinated development of energy and the environment through the development of renewable energy and efficient clean coal power [20]. Irene Monasterolo et al. developed a novel climate stress test methodology for portfolios of loans to energy infrastructure projects, by estimating the climate policy risks and non-systematic risks faced by China's overseas energy investment projects. They finally found that coal projects were greatly impacted by climate risks. The scholars suggested that China's policy banks should transform their investment to the green energy sector [21].

On the whole, most of the existing studies have examined the coal power investment risks in "Belt and Road" countries at the country level. Some scholars have analyzed the impact of overseas coal power on the environment. A few scholars have studied the impact of climate change on the Chinese overseas coal power projects. However, few works have studied the impact of various environmental risks on the economy of coal power projects invested by Chinese companies.

(2) The Role of Financial Institutions

Coal power investment is inseparable from the support of financial institutions. In recent years, under the trend of coping with global climate change, promoting world energy transformation, and low-carbon development, more and more financial institutions have announced their withdrawal from coal and electricity investment (Table 1). Research by IEEFA shows that so far, more than 100 major global financial institutions have withdrawn from the thermal coal sector. Every month since January 2018, a bank or insurance institution has announced its withdrawal from investment in coal mines and/or coal-fired power plants, and every two weeks, a financial institution that has announced an exit/closure policy has tightened its policy to close "loopholes" [22].

**Table 1.** Financing policies of financial institutions on the coal industry.

| Time | Financial Institution | Terms or Statements | Source |
|------|----------------------|---------------------|--------|
| 2013 | World Bank | Limit investment in coal power plants. | [7] |
| August 2013 | European Investment Bank (EIB) | European Banks would stop financing coal-fired power projects to help member states meet emissions targets. | [23] |
| 2015 | Organization for Economic Co-operation and Development (OECD) | The 29 member states agreed to limit financing for coal power plants through policy changes by the export credit agency. | [7] |
| September 2015; March 2016 | - | The two sides pledged to priorities funding and encourage the gradual adoption of low-carbon technologies using public resources. | [24] |
| July 2018 | Asian Development Bank (ADB) | After funding a supercritical coal power project in Pakistan in 2013, the ADB has yet to support any coal power projects. | [25] |
| October 2018 | World Bank | The World Bank exited the Kosovo coal-fired power plant project and terminated the investment in coal power plants. | [26] |
| December 2018 | European Bank for Reconstruction and Development (EBRD) | EBRD no longer supports coal mining and coal-fired power projects, including upgrading existing power plants or building new ones. | [26] |

However, Chinese finance has increasingly become a lender of last resort to coal-fired power plants as other banks take aggressive steps to limit their funding [6]. A survey of international coal financing by state-owned policy banks found that China is by far the largest supporter of future coal plants abroad with 44 GW of capacity, followed by South Korea with 14 GW, and Japan with 10 GW [27]. In 2018, Chinese financial institutions provided about $36 billion overseas for coal power

projects [28]. While the Chinese government has claimed that it will restrict coal lending, it has not formally restricted investment in coal-fired power plants. China's overseas energy investment model is more like a "portfolio of investments" involving multiple institutions and departments including the China Development Bank (CDB), overseas cooperative financial institutions, and Chinese companies and local implementors. CDB's energy investments account for a high proportion of its overall overseas lending, mainly in coal-related industries, while new energy accounts for a relatively small proportion [29].

There is no denying that China's financial institutions have also made efforts to limit the development of coal power in recent years. China has responded positively to the issue of limiting investment and financing in the coal industry, which is highly polluting and energy-consuming. For example, India's Adani Group has proposed the Carmichael coal mine project in the Galilee Basin in Queensland, Australia, and after considering the huge environmental costs, China's three banks (China Construction Bank, Industrial and Commercial Bank of China, Bank of China) issued a statement that said that they would not give financial support to the Carmichael coal mine project [26].

Facing the controversy of overseas coal power projects investment, it is necessary for Chinese financial institutions to incorporate the environmental risks of overseas coal power projects into their investment decisions. However, there is a lack of specific quantitative risk tools.

(3) Stress Test Method

Since the 1990s, many scholars have used the approach of stress testing, primarily in the financial sector. Merton incorporated the fluctuations of financial institution asset prices into the measurement model of default probability, which improved the prediction accuracy of default probability [30]. Wilson further optimized Merton's predicting model and established a stress test model specific to the credit risk default probabilities of commercial banks and major macroeconomic variables [31]. Blashke et al. introduced the basic framework of stress testing and invented a simplified method for processing data in financial systems [32]. Based on the market risk model, Alexander and Sheedy proposed a new method and used it to compare the performance of eight risk models in different rolling estimation periods [33]. Ba and Zhu discussed the practical implementation of the stress-testing model in developing countries where data are lacking [34]. Xu and Liu conducted a comparative analysis of several typical stress test systems and provided policy recommendations for China to assess the stability of the financial system [35]. At present, the stress test is widely used in the financial field and has become one of the most important tools for risk measurement and management. Stress testing was then gradually applied to other industries, and studies related to the coal-fired power industry have also emerged. In 2015, the Industrial and Commercial Bank of China (ICBC) conducted stress tests on two key polluting industries, thermal power and cement. In September 2017, Trucost assessed the potential environmental risks in China by focusing on the coal chemical industry [36]. In addition, Yuan and Wu conducted research on the application of stress testing to the development of coal-fired power in China. By establishing a complete stress test framework and a coal-electricity enterprise risk change stress test conduction map, the researchers examined the influence of energy efficiency standards, pollutant discharge tax, carbon market, water resource tax, overcapacity, and other renewable energy consumption factors on the value of coal-fired generating units [37].

These studies have laid a foundation for applying the stress test to study the environmental risks of coal power projects.

The contribution of this paper is twofold. First, it identifies the risk factors related to China's overseas coal power investment and presents a stress testing framework. Second, it examines the sensitivity of coal project value in two case countries from the perspective of environmental risks.

## 3. Environmental Stress Test

China's overseas investment in coal power will face country risk [12,13], power regulatory risk, climate risks [21], and natural environmental risk. In order to quantify the impact of these risks

on the economy of coal power projects, this paper selected the environmental stress test method. The environmental stress test is an approach to internalize environmental costs into corporate costs and measure the impact of environmental factors on the value of corporate assets. The traditional stress test, primarily used in financial institutions and banks, mainly includes the following steps: selecting pressure objects and determining pressure indicators, selecting pressure factors, constructing pressure conduction models, setting stress test scenarios, performing stress tests, and analyzing the results. The pressure-bearing object refers to the main body to be tested, that is, the coal-fired power project invested and constructed by the electric power enterprise in the "Belt and Road" countries. The pressure indicators refer to the observation index of the stress test subject when applying environmental pressure. Here, we set the enterprise value and the IRR as the pressure indicators.

This paper examined: (a) the direct environmental impact of coal-fired power projects; (b) the impact of changes in environmental risk data on the financial status of enterprises through the discounted free cash flow method; (c) changes in main business costs and income under the stress scenario; (d) main indicators of cash flow statement and balance sheet according to the hook relationship (the relationship between the relevant figures in financial statements, which can be used for mutual examination and verification) of financial statements and basic processing norms; and finally, (e) the new financial statement.

### 3.1. Discounted Free Cash Flow Method

Enterprise value is the market evaluation of the sum of the tangible assets and intangible assets of an enterprise. It is also an assessment of the profitability and equity value of the enterprise in the future. The current internationally-accepted assessment methods are mainly divided into three categories: income method, cost method, and market law. This paper used the weighted cost method to evaluate the enterprise value. Based on the assessment of a company's annual free cash flow in the future, and the weighted average cost of capital (WACC) as the discount rate, the present value of the discount is the enterprise value, as shown in Figure 1.

$$
\begin{aligned}
Corporate\ cash\ flow(FCFF) &= Earnings\ Before\ Interest\ and\ Taxes \times\ (1 - Income\ tax\ rate) \\
&+ Depreciation - Capital\ expenditures - Increased\ working\ capital \\
&= Net\ profit\ after\ tax - Net\ investment
\end{aligned}
\tag{1}
$$

$$
WACC = K_e \times W_e + K_d \times W_d \times (1 - T)
\tag{2}
$$

where $K_e$ is the cost of equity capital of the enterprise; $W_e$ is the proportion of corporate equity capital cost in the capital structure under market value; $K_d$ is the debt cost of the enterprise; $W_d$ is the proportion of corporate debt capital cost in the capital structure in the market value; and $T$ is the corporate income tax.

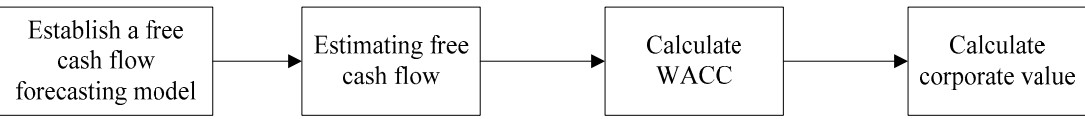

**Figure 1.** Estimation process of the enterprise value.

Among them, the cost of equity capital is determined by the capital asset pricing model (CAPM), which reflects the higher return required by investors for the increased risk through the adjusted equity risk premium of the $\beta$ coefficient, as shown in Equation (3).

$$
K_e = R_f + \beta \times (\overline{R}_m - R_f)
\tag{3}
$$

where $R_f$ is the risk-free rate of return; $\overline{R}_m$ is the market expected rate of return; and $\beta$ is the degree of response of a stock's rate of return to changes in market yield.

$$V = \sum_{t=1}^{n} \frac{FCFF}{(1+r)^t} \tag{4}$$

where $V$ is the enterprise value; $n$ is the life of the asset; *FCFF* is the corporate cash flow of the $t$ period; and $r$ is the discount rate.

## 3.2. Environmental Risk Analysis

The environmental risks studied in this paper are not only related to the natural environment, but also related to the political and financial environments.

In recent years, more and more countries have separated power generation from power grids and have used independent power producer (IPP) models to develop energy projects in order to improve the efficiency of power market operations and the utilization of domestic and foreign private capital [38]. Chinese power companies also use this model when investing in coal-fired power projects overseas. Most IPP projects are funded by financing. At present, the main financing channels include the Export–Import Bank of China, China Development Bank, or the signing of loan agreements between countries [39]. According to the asset value of the project and the cash flow generated by it, a special project company is established for financing, construction, and operation. The process of coal power projects involves government departments, investors (shareholders), power purchasers, lending banks, insurance agencies, engineering contractors, operations, and many other relevant parties. A series of agreements and contracts are signed between these project participants to determine the arrangement of risk sharing, clarify their rights and obligations, and form a close cooperative relationship (Figure 2).

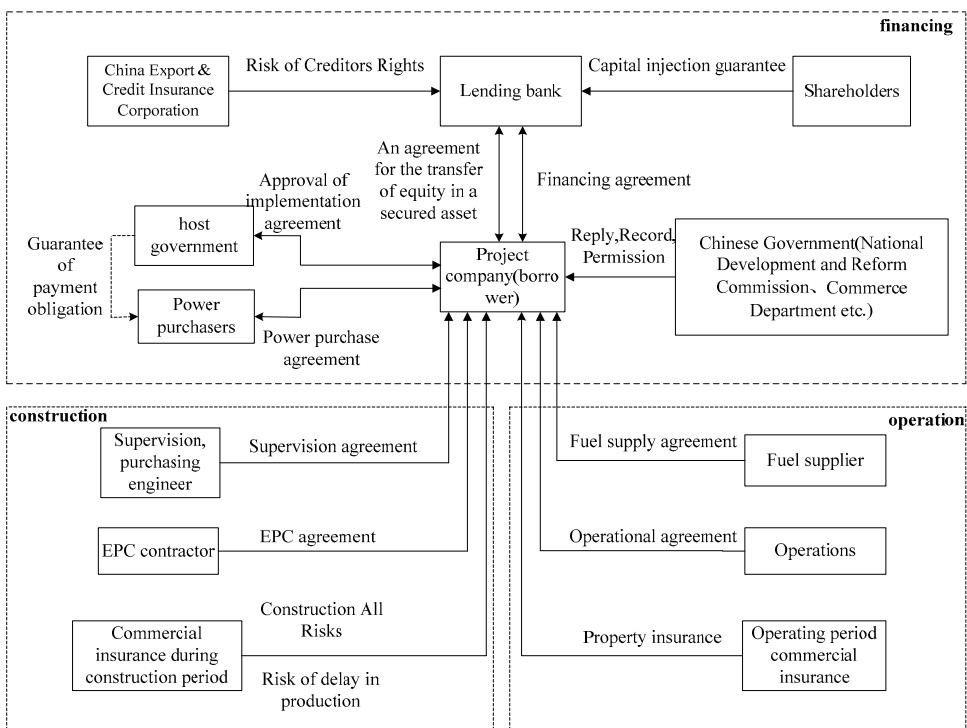

**Figure 2.** Major participants and their agreement relationship in coal power projects.

For the project company, the risks involved in the construction phase are related to the technical level and management level of the enterprise itself, which are not affected by the environment of "Belt and Road" countries, so these risks were not considered.

In the operation phase, power enterprises will face fuel supply risk and coal power planning risk. The fuel supply risk will be transmitted to fuel cost through coal price, affecting the operating cost of the power plant. Coal power planning risk affects the operating income through the utilization hours.

In the financing process, the project company will sign a power purchase agreement (PPA) with the purchaser (mostly a power company with a government background). It is one of the key agreements that need to be negotiated. The PPA core clause clearly stipulates the design of the electricity price mechanism, electricity purchase obligation of the electricity purchaser, power supply obligation of the power seller, operation, and maintenance of the power plant, coal supply responsibility and risk, government guarantee, and force majeure [40]. These terms will directly affect the risks, benefits, and guarantees of Chinese power companies during construction and operation. Since electricity is often purchased in local currency, fluctuations of the local currency exchange rate will be transmitted to the operating income through the electricity price. Second, the coal power project will need to be approved by the local government before implementation. Faced with the pressure of environmental protection and NDC goals, the local government will strengthen its environmental supervision, thus increasing the environmental protection cost of the coal power project.

### 3.3. Stress Testing Conduction Path

A stress test conduction model is at the heart of environmental stress testing. Figure 3 shows that the model takes into account the influence of environmental risks on the main business income and expenditure of coal-fired power projects. It includes five pressure factors: coal price, utilization hours, exchange rate, carbon tax, and environmental protection requirements. This paper simulated the impact of various stress factors on the enterprise value and IRR (Appendix A).

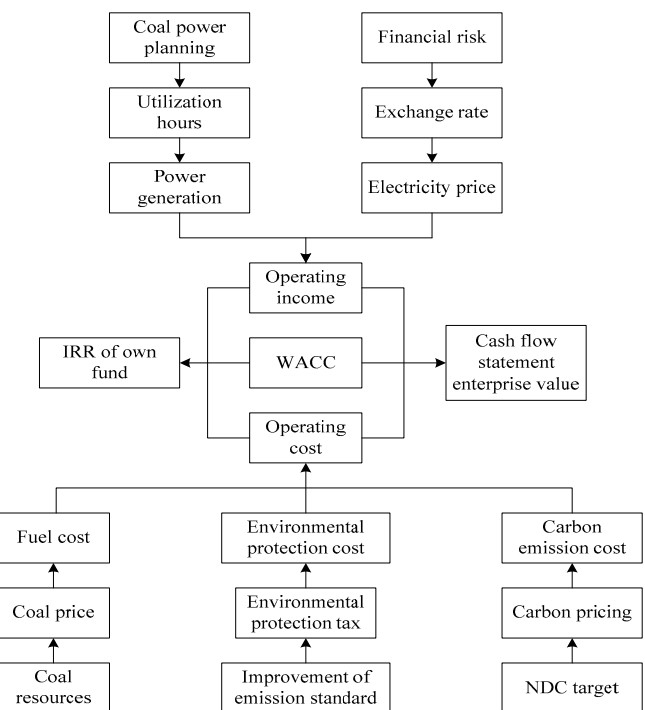

**Figure 3.** Environmental risk pressure conduction map of a coal power project.

## 4. Cases and Scenario Setting

In recent years, Indonesia and Vietnam have become the hot countries for China to invest in coal power projects, which is the main reason for this paper choosing these two countries as the focus of study. South Asia and Southeast Asia are the two primary regions for China's overseas investment in coal power projects, which can be attributed to their relatively stable political environment, fast-growing

economy, and geographical proximity to China and accounted for 57.11% and 22.75%, respectively, of China's total coal power installations in "Belt and Road" countries. The coal power investment in South Asia was mainly concentrated in India, accounting for 90.35% of the region, but China's participation in India's coal power is mostly through equipment export, and the investment in coal power projects is mainly concentrated in Southeast Asia. Moreover, due to policy changes in India and the economic development in Southeast Asia after 2010, China's participation in coal power generation in South Asia has gradually decreased, and participation in Southeast Asia continues to increase. By the end of 2016, Indonesia and Vietnam were the first and second largest installed capacity countries in Southeast Asia, respectively [41] (Figures 4 and 5), and Chinese finance supports 30% of all coal-fired capacity under development in Vietnam and 23% in Indonesia [7].

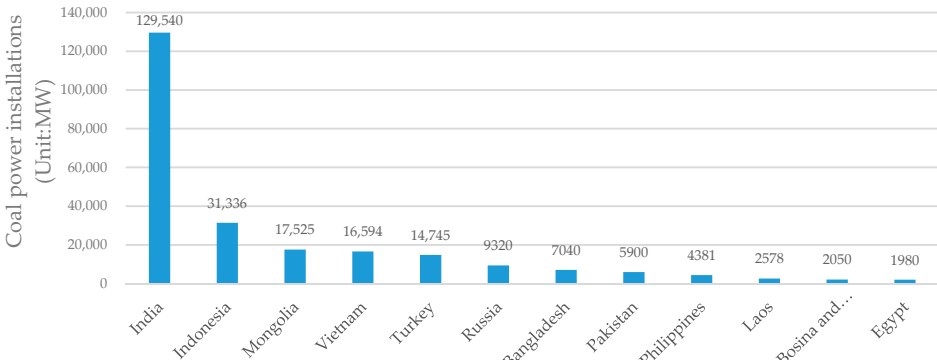

**Figure 4.** China's participation in coal power installations in some "Belt and Road" countries. Data sources: Institute of Global Environment, A Survey of China's Participation in Coal Power Projects in the "Belt and Road". Note: The figures show all of the coal power projects China has participated in including those in operation, construction, signed, planned, shelved, and cancelled.

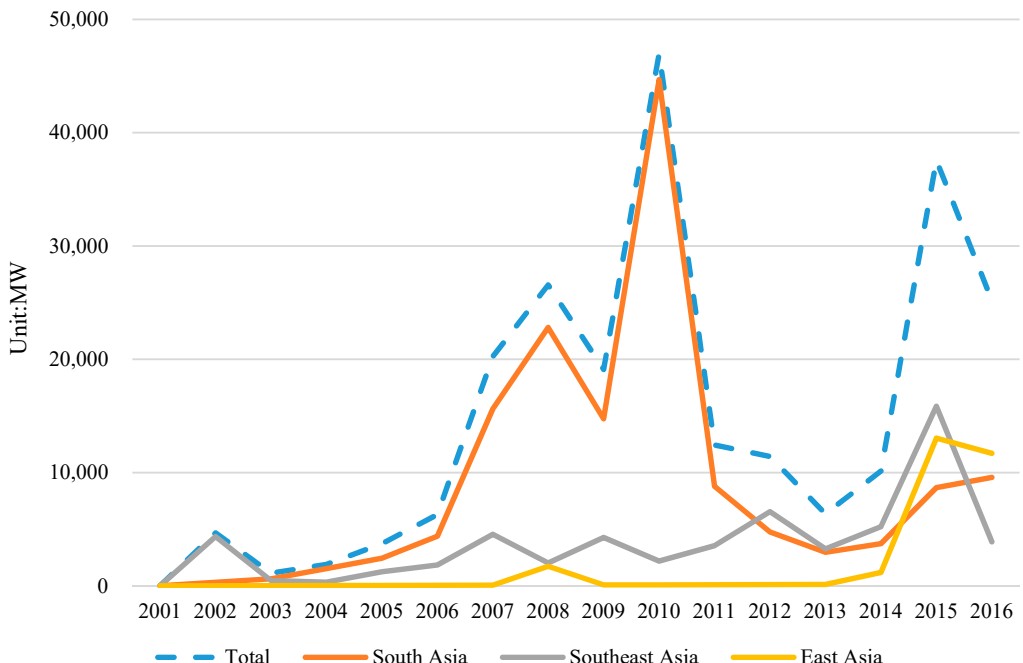

**Figure 5.** China's annual participation in various regions of coal power project installation and total installation. Data sources: Institute of Global Environment, A Survey of China's Participation in Coal Power Projects in the "Belt and Road". Note: The data in the figure are the installed capacity of new projects in that year. New projects are those signed or started during the year.

In addition, from the present study of power development in Indonesia and Vietnam, coal power in these two countries will experience big developments and attract great investment from China due to vast market demand and national policy support. Wang and Xu focused on the electricity market in Indonesia including power installed capacity and electricity consumption, grid status, power planning, electricity price, and the Indonesian State Power Corporation (PLN) operation status [42]. Wang et al. analyzed Indonesia's power investment environment and the investment prospects of various power stations [43]. Utama and Ishihara and others predicted the future power demand trends in Indonesia [44]. Liu et al. [45] and Niu et al. [46] pointed out that Indonesia is one of the key overseas resource-target areas for Chinese coal-fired power companies. Tran-Quoc et al. proposed a method to improve voltage stability through downconverter technology in Vietnam [47]. Finenko A and Thomson E proposed that coal power would become the most important source of electricity in Vietnam in the next 10 years [48] and Sebastian et al. expected Vietnam's power installed capacity to increase five-fold from 2013 to 2030 [49].

*4.1. Stress Test Factors*

Due to the different risk factors, coal power projects in Indonesia and Vietnam face different environmental risks (Table 2).

**Table 2.** Environmental risks faced by coal power projects.

| Environmental Risks | Indonesia | Vietnam |
| --- | --- | --- |
| Coal resources | Rich in coal resources but low in quality. | Coal dependence on imports. |
| Coal power planning | Electricity growth slows down, coal power will overcapacity. | Renewable energy development, coal power will overcapacity. |
| Financial risk | Exchange rate fluctuation. | Exchange rate stability. |
| Environmental protection standard | Improvement of emission standard. | |
| NDC target | Pressure to achieve NDC goals. | |

The following is a detailed analysis of these risks in terms of the five stress test factors.

(1) Coal price

Changes in coal prices directly affect the fuel cost of the plant, which will cause changes in enterprise value and IRR. Coal price is influenced by the richness of the coal resources, the quality of the coal, and the mining conditions where the coal power projects are located.

Indonesia is a coal exporter with abundant coal resources. By 2017, the reserves of coal resources were about 22.6 billion tons, accounting for 2.2% of the world's total reserves. The reserve-production ratio was 49 [50]. The reserves of coal are mainly distributed in the two islands of Sumatra and Kalimantan. Indonesia has a complete range of coal storage, mainly lignite, sub-bituminous coal, bituminous coal, and anthracite. The coal metamorphism is graded from medium to low, with high moisture, low ash (usually less than 10%), low sulfur (usually less than 1%), and high volatility. In 2017, Indonesia's coal output was 407 million tons, and domestic coal consumption was 85.8 million tons, accounting for 32.63% of primary energy consumption (BP Statistical Review of World Energy 2018). Most of the coal mines are open pit mines. However, with the increase of mining volume, open pit mines will gradually decrease, as will the difficulty of future mining, so coal prices will rise slightly.

Compared with Indonesia, Vietnam's coal resources are more inadequate. By the end of 2017, the reserves of coal were only 3.36 billion tons including hard coal, lignite, and peat. In 2017, Vietnam's coal production was 31.95 million tons, basically thermal coal. Coal consumption was 42.3 million tons, and 39% of the electricity was supplied by coal. However, Vietnam's power planning goal is that by 2020, the total installed capacity of coal-fired power reaches 26 gigawatts, and the power generation reaches 131 billion GWh, accounting for 49.3% of all types of electrical energy. According to

the Green Innovation and Development Center (Green ID), Vietnam's coal power generation will reach 137 million tons of standard coal in 2030, with a coal import ratio of 75.7% (Figure 6) [51]. The expansion of the scale of coal power development in Vietnam will increase its reliance on imported coal, which will increase coal prices.

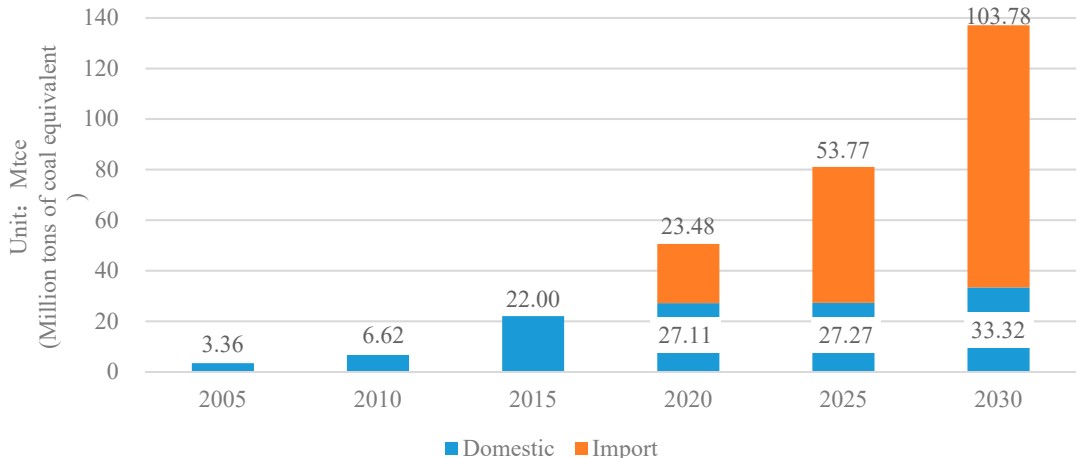

**Figure 6.** Coal source for power generation in Vietnam. Data sources: Green ID "Analysis of future generation capacity scenarios for Vietnam".

(2) Utilization hours

Coal-fired planning risks include three areas: coal-electricity construction economics, coal-fired power adequacy, and resource constraints. Investment in coal-fired power projects in the "Belt and Road" countries needs to consider the feasibility of project construction, the economics of operation, and the power demand and supply. This paper focused specifically on the economics of project operation, and the impact of the utilization hours of coal-fired projects based on the power installation plan of the "Belt and Road" countries.

Recent years have seen an improvement in Indonesia's investment environment and a growth rate of around 5% in domestic economics. Additionally, with increases in population and electrification, the demand for electricity has continued to increase. According to the forecast of Perusahaan Listrik Negara (PLN), Indonesia's electricity demand will grow at an average annual rate of more than 7.5% from 2015 to 2025, and power users will increase by 21.7 million (Figure 7) [52]. Coal power development ranks as one of the top in Indonesia's power development plan, while coal-fired power generation accounts for more than half of the total power generation. As of April 2017, Indonesia's coal power has developed 23,345 MW, accounting for 51.2% of total installed capacity. The 2016–2025 plan states that by 2025, the newly installed capacity of coal-fired power stations will be 34,800 MW, accounting for 43% of new installed capacity. However, Indonesia's coal resources are unevenly distributed. The cost of coal-fired power generations is lower and the scale of generation is more expanded in resource-rich areas. However, with the slowdown in the growth of power demand and the development of renewable energy, there will be overcapacity risks in coal-fired power projects in these areas, which will also result in a decrease in utilization hours.

Like Indonesia, the recent years have seen continued growth in electricity demand and expanded installed power capacity in Vietnam. In 2016, Vietnam's total installed capacity of electricity reached 41.29 GW, of which coal-fired installed capacity reached up to 14.44 GW, accounting for 35%. Coal-fired power has become the main new power source in recent years (Figure 8). However, unlike Indonesia, Vietnam's coal is mostly imported, which results in higher risks. Vietnam is vigorously developing renewable energy such as wind, solar and biomass in order to protect national energy security and respond to changes in the global climate. If it succeeds in finding a suitable alternative energy source,

its dependence on coal power will be reduced. Then, coal-fired power projects may run into the risks of overcapacity, and the utilization hours will also be decreased.

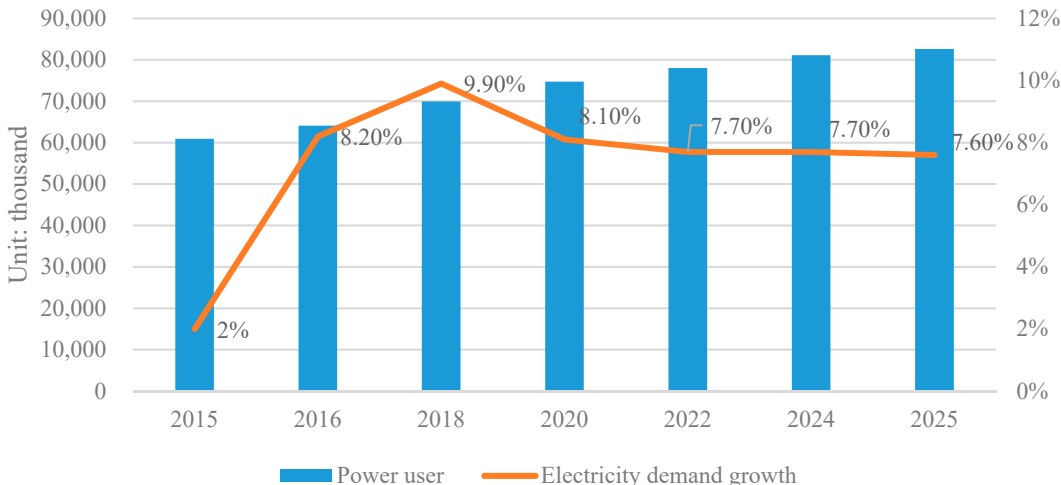

**Figure 7.** Indonesian power growth forecast for 2015–2025. Data sources: Zhengdian International "Overview of Indonesia's electricity market and investment prospects".

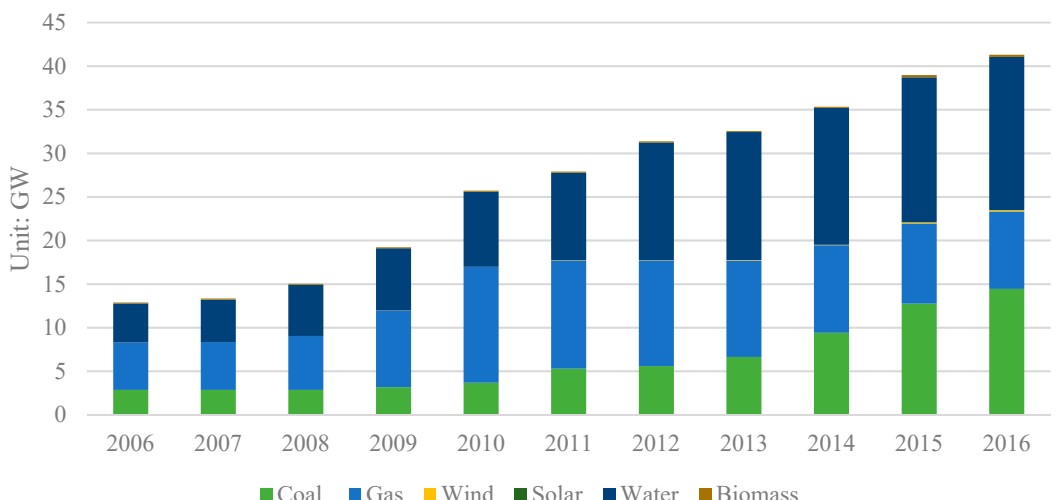

**Figure 8.** Electricity installations in Vietnam from 2006 to 2016. Data sources: The electricity installations in Vietnam between 2006 and 2016 were calculated by integrating data from the International Energy Agency (IEA), the International Renewable Energy Agency (IRENA), and the Coal Plant Tracker and Platts.

(3) Exchange rate

The transaction of electricity is conducted in local currency for coal power projects developed by China in the "Belt and Road" countries. Therefore, the value of the local currency directly affects the revenue of enterprises. The exchange rate of the Indonesian rupiah (IDR) is highly volatile. Due to the Asian financial crisis in 1998, the U.S. dollar (USD) relative to the IDR rose to 12,000. After that, the global financial crisis in 2008 caused the exchange rate to rise to 12,300 [53]. In 2018, the Federal Reserve continued interest rate hike and the sharp depreciation in the currencies of emerging economies resulted in another sharp depreciation in the IDR. On October 2, the exchange rate of the Indonesian rupiah relative to the USD relative to the IDR reached 15,048, which rose above 15,000 for the first time [54]. Affected by several economic fluctuations, the exchange rate of the Chinese Yuan (CNY) relative to the IDR has also experienced great fluctuations since the beginning of the 21st century (Figure 9) [55].

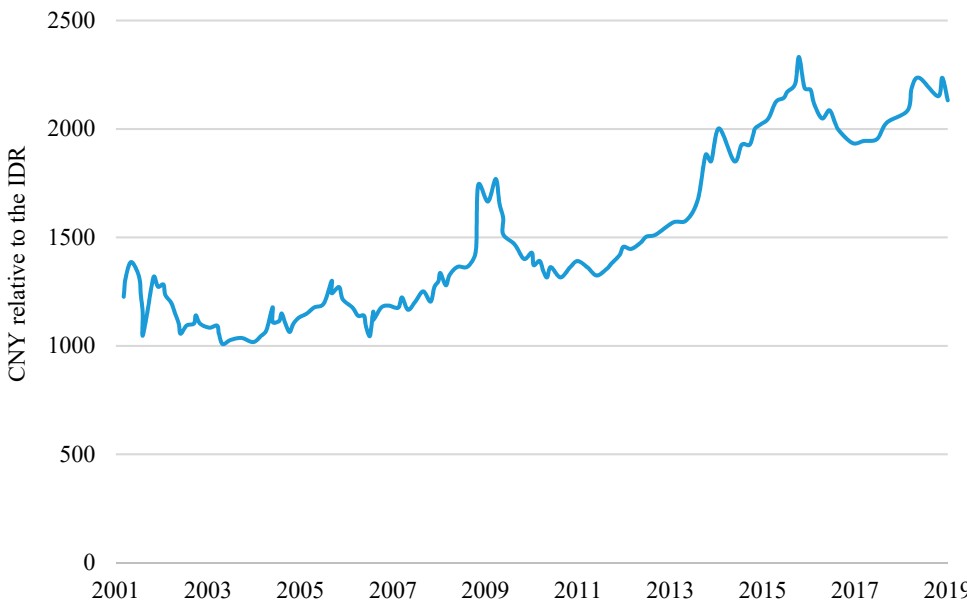

**Figure 9.** Trends of the CNY relative to the IDR in 2001–2019. Data sources: Yahoo Finance. Currency Converter.

Despite the value of the Vietnamese Dong (VND) being relatively stable, it faces the risk of currency depreciation because Vietnam's exports account for a high proportion of GDP (gross domestic product) (more than twice that of Indonesia) and some merchants tend to make the VND exchange rate lower. At present, the exchange rate of the USD relative to the VND has remained at around 3500 (Figure 10).

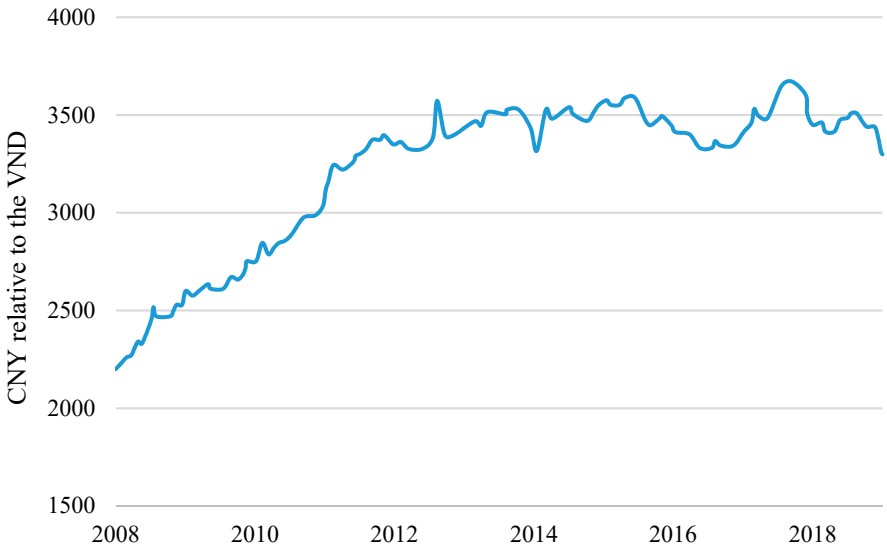

**Figure 10.** Trends of the CNY relative to the VND in 2008–2019. Data sources: Yahoo Finance. Currency Converter.

(4) Environmental requirements

In operation, coal-fired power plants generate pollutants including nitrogen, sulfur-containing substances, and soot. As shown in Table 3, many countries have regulated the limits of the pollutant discharge of coal-fired power plants. However, Indonesia, Vietnam, and other "Belt and Road" countries have lower pollutant-discharge standards compared to China and Western developed

countries. As coal-fired power projects in these countries at present do not have any problem of environmental taxes, the company can obtain higher returns with lower protection expenses.

**Table 3.** Comparison of current air pollutant emissions limiting standards in major coal power countries in the world [56].

| Country | $SO_2$ (mg/m$^3$) | | NOx (mg/m$^3$) | | PM (mg/m$^3$) | |
|---|---|---|---|---|---|---|
| | Active | New | Active | New | Active | New |
| China | 200–400 | 50 | 200 | 35 | 30 | 15 |
| EU | 200–400 | 150–400 | 200–450 | 150–400 | 20–30 | 10–20 |
| America | 160–640 | 160 | 117–640 | 117 | 23 | 23 |
| India | 200–600 | 100 | 300–600 | 100 | 50–100 | 30 |
| Indonesia | 750 | 750 | 850 | 750 | 150 | 100 |
| Japan | - | - | 123–513 | 123–513 | 30–100 | 30–100 |
| Philippines | 1000 | 200 | 1000 | 500 | 150 | 150 |
| Korea | 286 | 229 | 308 | 164 | 40 | 20–30 |
| Thailand | 700 | 180 | 400 | 200 | 80–320 | 80 |
| Vietnam | 1500 | 500 | 1000 | 650 | 400 | 200 |

Data sources: IEA "World Energy Outlook Special Report 2016 Energy and Air Pollution".

In response to environmental pressure and NDC commitments, Indonesia and Vietnam will raise their environmental standards and impose environmental taxes in future. The coal-fired power projects should not only raise their own technical standards, but also increase expenditure on environmental protection and taxation. However, this will increase the project's expenses. In this case, these countries need to provide electricity price subsidies to the coal-fired power project, otherwise, the enterprises will have to take on more financial burdens.

(5) Carbon tax

On 5 October 2016, about 200 countries jointly adopted the Paris Agreement at the Paris Climate Conference and formulated an institutional arrangement to deal with climate change. The purpose was to control the global average temperature increase within 2 °C in this century and strive for 1.5 °C. To this end, all countries have set their own NDC goals. By 2030, these countries will raise the carbon emission standards for coal-fired power projects in order to fulfill their goals. Power companies must seek strategies to deal with carbon and environmental risks, which will bring about increased costs and reduced profits due to the addition of environmental taxes.

Indonesia's NDC goal is to control greenhouse gas emissions within 2.881 billion tons and to reduce it by 29% in 2030 under the current scenario BAU (Business As Usual). It can be inferred from the carbon intensity of coal-fired power in Indonesia in 2015 that the carbon emissions of coal-fired power in Indonesia will be around 330 million tons at that time. If Indonesia improves the technical level and energy efficiency of the coal power plants, it will reduce the carbon emission intensity of coal-fired power to 850 g/kWh. In 2030, the carbon emissions of coal-fired power will be 264 million tons, which will be 0.66 billion tons lower. The contribution rate to the unconditional reduction of the 29% target is 21%, and the contribution rate to the conditional reduction of 41% target is 16.6% (Figure 11) [57]. According to the findings of IRENA's study, in 2030, Indonesia's total carbon emissions of energy-related industries (electricity, industry, transportation, and construction) in the BAU scenario will be 1.253 billion tons, and the carbon emissions in power industry will be 605 million tons, among which the carbon emissions of coal and electricity will be 330 million tons (estimated results), accounting for 26.3% of the total carbon emissions of energy-related industries. Comparing the proportion of coal-fired carbon emissions and the contribution rate of emissions reduction, the reduction of coal-fired carbon emission intensity to 850 g/kWh is still insufficient to complete the emissions reduction task.

Vietnam's NDC goal is to reduce the carbon intensity of GDP by 20% in 2030 compared to 2010, with the ultimate goal of 30%. The carbon intensity of GDP has remained around 0.9 kg/USD in recent

years (Figure 12). Assuming that Vietnam can maintain a growth rate of 6.8% and achieve the goal of reducing carbon intensity by 20% by 2030, Vietnam needs to obtain 526 billion yuan of GDP with 625 million tons of emissions in 2030. This is difficult for Vietnam if it continues to develop coal power.

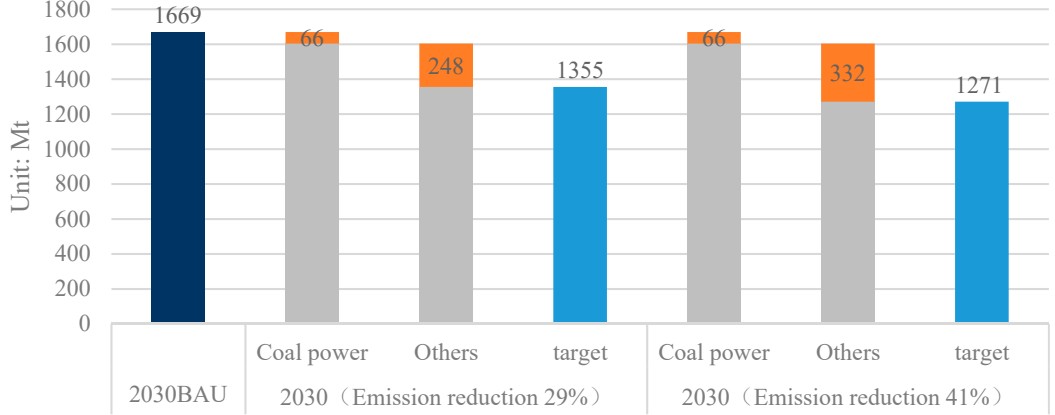

**Figure 11.** Indonesia's greenhouse gas emissions reduction scenario in the energy sector. Data sources: IRENA' Renewable Energy Policies in a Time of Transition'.

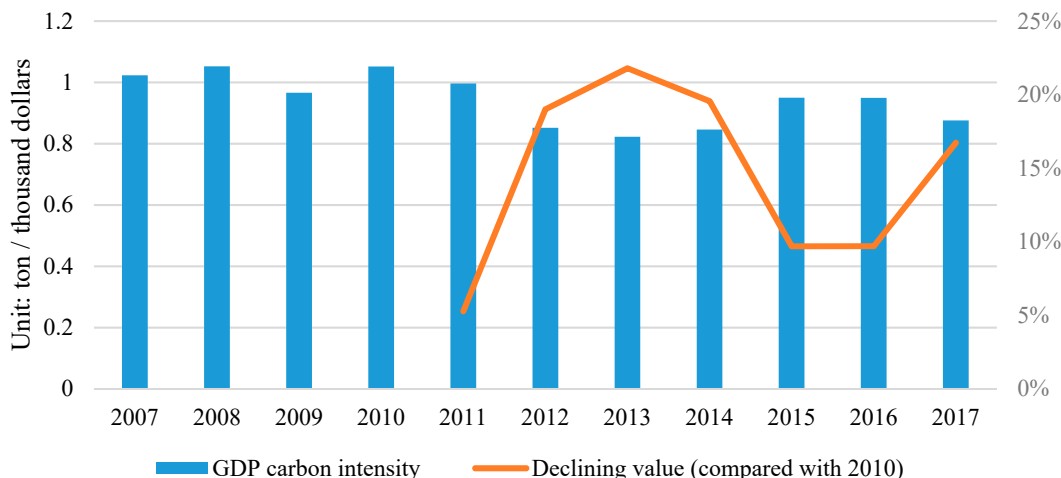

**Figure 12.** 2007–2017 Vietnam's GDP carbon intensity changes. Data sources: BP. Statistical Review of World Energy 2018.

*4.2. Stress Test Scenario*

Before setting the stress test scenario, we first calculated the enterprise value and internal rate of return of the average level of coal-power investment projects in these two countries and selected two typical coal power projects (the GH EMM Indonesia Project and Vietnam Vinhtan Coal Power Project) for comparative analysis. The GH EMM Indonesia Power Plant is the first coal-electricity integration project invested by the Shenhua Group overseas [58]. In 2017, it was successfully awarded three awards for Indonesia's "Five Best Power Enterprises", "Five Best Innovative Power Enterprises", and "Five Best 100 MW Power Enterprises", and won the "Best Innovative Power Enterprise of 2017" for Indonesia. The Vietnam Vinhtan coal-fired power plant is the largest coal power project invested by Chinese enterprises in Vietnam [59]. It is also a key production capacity cooperation project in the five-year development plan of the China–Vietnam economic and trade cooperation and the five-year plan of onshore infrastructure cooperation [60]. The data under the relevant reference value are shown in Table 4.

**Table 4.** Related parameter setting of coal power project.

| Parameter | Unit | Indonesia | | Vietnam | |
|---|---|---|---|---|---|
| | | Average | GH EMM | Average | Vietnam Vinhtan |
| Installed capacity | MW | 300 * | 300 | 1200 * | 1200 |
| Expected utilization hours | Hours/year | 5396.61 | 5667 | 5000 | 6500 |
| Coal consumption rate | thousand standard coal per MWh | 330 | 360 | 298 | 306 |
| Coal price | yuan/ton | 586.5 | 70 | 614 | 440 |
| Calorific value of coal | kilocalorie | 7000 | 2376 | 7000 | 5000 |
| Unit kilowatt investment cost | yuan/kW | 9450 | 8390 | 11,030.42 | 10,091.25 |
| Self-use rate | % | 10.2 | 10.2 | 5 | 5 |
| Plant life | year | 30 | 30 | 25 | 25 |
| Discount period | year | 16 | 16 | 15 | 15 |
| Exchange rate | CNY relative to IDR(VND) | 2011 | 2011 | 3303 | 3303 |
| Enterprise value | yuan | 5,436,718,981 | 5,532,164,503 | 25,285,271,864 | 23,547,969,913 |
| IRR of own funds | - | 13.95% | 14.69% | 12.35% | 11.67% |

* 300 MW or 2*600 MW are designed to facilitate comparison with two specific coal-power projects in Indonesia and Vietnam. Data sources: The data of the average level are provided in Appendix B. Relevant data of two coal power projects were obtained from field research in April 2018, and data of the same period were selected for the exchange rate. In addition, the exchange rates of the two countries were collected at the beginning of the survey. At that time, the two countries did not have any planned tax on carbon and environmental risks thus, they did not consider the benchmark value.

The results show that coal power investment in Indonesia has a relatively high profit state: the enterprise value was 5.436 billion yuan and the IRR was 13.95%. The GH EMM power plant had a low cost of coal (uses inferior lignite), so the overall income was higher than the average level of Indonesia's coal power investment. The enterprise value of coal power in Vietnam was calculated to be 25.285 billion yuan by using the discounted free cash flow approach, and the IRR was 12.35%. The Vietnam Vinhtan power plant's revenue was in line with the average level.

Next, we examined the capacity of the power plants to withstand environmental risks. We estimated how the five stress factors influenced the corporate value and internal rate of return in both optimistic and pessimistic scenarios.

Table 5 presents the values set for the five stress factors on both optimistic and pessimistic scenarios. (1) Coal price. Indonesia is located in coal-rich areas, thus has no risk of change in coal price, so we set some fluctuations based on Indonesian coal prices in recent years. Vietnam's coal relies on imports, so the coal price of Vietnam may increase. This paper selected the global average price of steam coal ($80 per ton) and assumed that its calorific value was 5500 kcal [61]. (2) Utilization hours. The changes in the utilization hours of the coal power plants in two countries are consistent: it will increase in recent years, but tend to decrease in the long run. (3) Exchange rate. Indonesia's exchange rate in the optimistic scenario was relatively stable, and the lowest value in recent years was set for the pessimistic scenario. Fluctuation in Vietnam's exchange rates is small, so a slight variation exists between the optimistic and pessimistic scenarios. (4) Carbon tax and environmental requirements. Indonesia and Vietnam have a certain gap with China in terms of power development, considering that they are both developing countries and are currently facing pressure to achieve the NDC goals. Under the pessimistic scenario, the carbon tax and environmental tax were set according to the pilot data of China's early carbon market, and assumed that the government will not subsidize desulphurization, denitrification, and dust removal. Under the optimistic scenario, the carbon price and environmental protection tax were appropriately reduced. At the same time, it was assumed that the government would provide some subsidies toward the cost of environmental protection. The data were also set according to the data of China's electricity market. Considering that the carbon market in the "Belt and Road" countries has just started, our consideration of carbon price and environmental protection tax was relatively low. For example, the cost of power generation in China has increased by nearly 10% from the relatively low emission standard to the current ultra-low emission standard [9]. However, a

direct assumption of 10% growth is not particularly realistic in the near to medium term for "Belt and Road" countries.

**Table 5.** Stress test scenario setting.

| Stress Factor | | Unit | Indonesia | | Vietnam | |
|---|---|---|---|---|---|---|
| | | | Optimistic | Pessimistic | Optimistic | Pessimistic |
| Coal price | | yuan per ton | 552 | 621 | 630 | 703 |
| Utilization hours | | hours per year | 5800 | 5000 | 5500 | 4800 |
| Exchange rate | | CNY relative to IDR(VND) | 1800 | 2330 | 3200 | 3500 |
| Carbon tax | Carbon price | yuan per ton | 10 | 30 | 10 | 30 |
| | Carbon price growth rate | % | 10 | 18 | 10 | 18 |
| | Carbon market paid quota ratio | % | 10 | 30 | 10 | 30 |
| Environmental requirements | Environmental tax $SO_2/NO_x$ | yuan per ton | 800 | 1200 | 800 | 1200 |
| | Environmental tax soot | yuan per ton | 200 | 250 | 200 | 250 |
| | Electricity price subsidy | yuan per kWh | 0.03 | 0 | 0.03 | 0 |

## 5. Empirical Results

With our approach, we examined how the five stress factors affected the corporate value and internal rate for both the optimistic and pessimistic scenarios. We calculated the corporate value and internal rate using the free cash flow discounting method, based on the values of five stress factors set for the two different scenarios. Finally, we compared them with the baseline values to obtain the stress test results. Pessimistic refers to the outcome of a pessimistic situation. Optimistic–pessimistic refers to the difference between the optimistic scenario and the pessimistic scenario. The average refers to the average level of Indonesia or Vietnam.GH EMM refers to the GH EMM Indonesia Project. Vietnam Vinhtan refers to the Vietnam Vinhtan Coal Power Project.

(1) Indonesia

The sensitivity of coal power investment projects in Indonesia to various risks is ordered as follows: exchange rate > environmental protection requirements > utilization hours > carbon price > coal price (Figures 13 and 14).

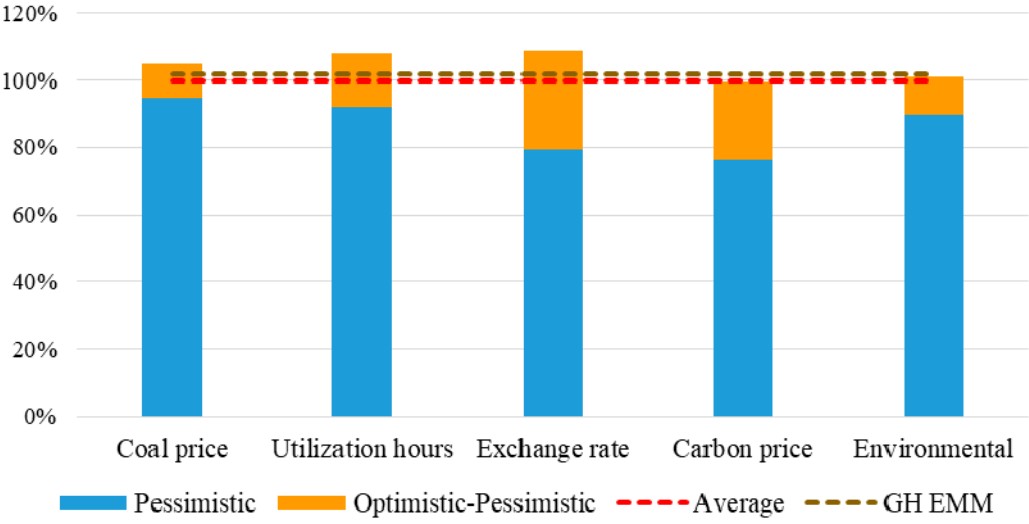

**Figure 13.** The Indonesia Coal Power Project's stress test results: enterprise value.

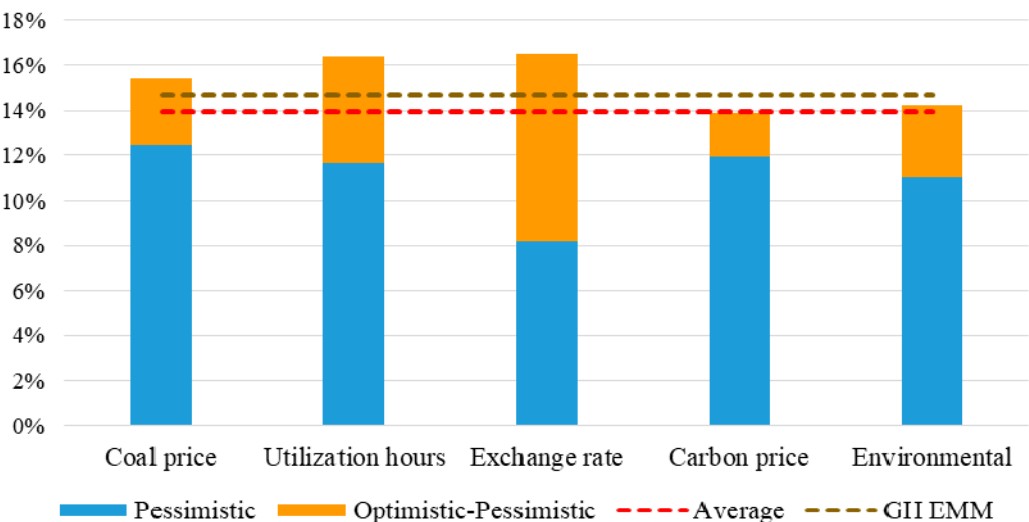

**Figure 14.** The Indonesia Coal Power Project's stress test results: IRR of own funds.

When the utilization hours increase to 5800 h/year and the exchange rate of the CNY relative to IDR falls to 1800, the enterprise value will increase by about 8%, and the IRR will increase by nearly 18%. When the utilization hours are reduced to 5000 h/year and the value of the CNY relative to IDR rises to 2330, the value of the enterprise is reduced by 7.86% and 20.7%, respectively, and the IRR is reduced to 11.68% and 8.16%, respectively. The impact of coal prices is relatively small. In the optimistic scenario, the value of the enterprise increased by 5.06%, and the IRR increased to 15.43%. In the pessimistic scenario, the enterprise value reached 94.94% of the benchmark value and the IRR decreased by 10.32%.

The profits of the Indonesia Coal Power Project will be greatly affected once it includes carbon the tax and environmental protection requirements in its costs. In the optimistic scenario, which is featured by a lower carbon price quota ratio, and environmental protection tax, environmental protection electricity price subsidies, the value of the company can be maintained within the baseline level with slight fluctuations. In the pessimistic scenarios, which feature a higher carbon price, environmental taxes, and zero subsidy, the enterprise value will fall by 23.9% and 10.28%, respectively, and the IRR will fall by about 2.5%.

Indonesia's abundance in coal resources and increase in power demand have provided Chinese power companies sound investment opportunities in coal-fired power projects. However, coal-fired power projects face many challenges: (1) With low-carbon transformation and the development of renewable energy, coal-fired power projects are likely to run into the risk of stranding assets. Additionally, the utilization hours will drop significantly, which has been shown to have a great impact on corporate value and IRR. (2) The exchange rate of the Indonesian rupiah fluctuates greatly, and the depreciation of the Indonesian rupiah will lead to a sharp decline in corporate income. (3) Chinese investors are likely to use advanced technologies for environmental protection, which have brought about an increase in costs. Chinese power companies tend to use efficient clean coal technology to reduce carbon dioxide and pollutants such as sulfur and nitrogen. If the Indonesian government increases the environmental taxes, but does not provide electricity price subsidies, the IRR will drop significantly.

(2) Vietnam

The sensitivity of coal power investment projects in Vietnam to various risks is ordered as follows: coal price > environmental protection requirements > carbon price > exchange rate > utilization hours (Figures 15 and 16).

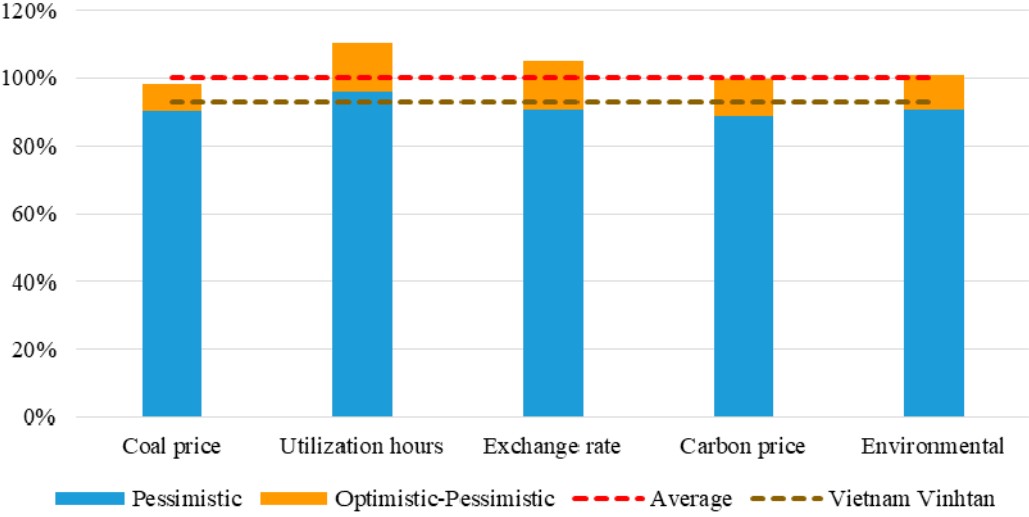

**Figure 15.** Vietnam Coal Power Project's stress test results: enterprise value.

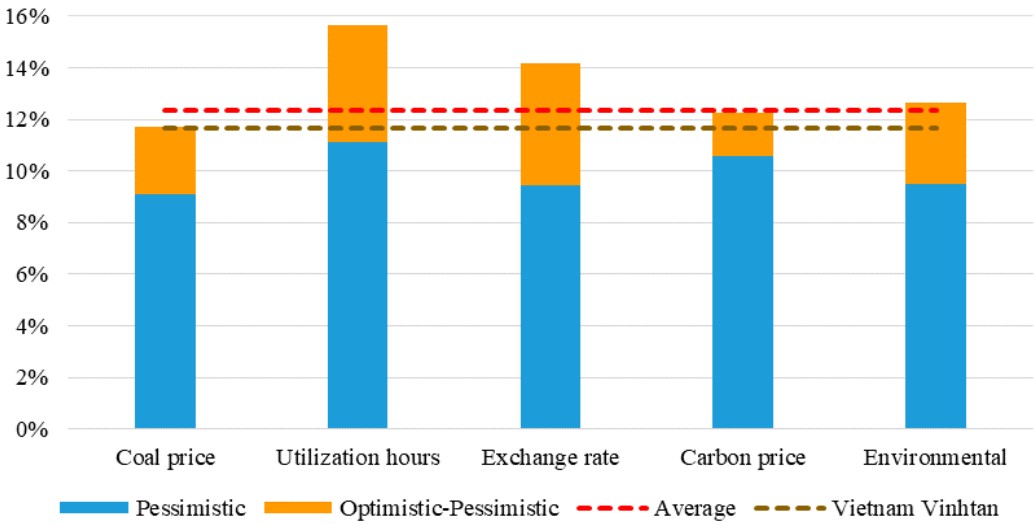

**Figure 16.** Vietnam Coal Power Project's stress test results: IRR of own funds.

In the pessimistic scenario, coal prices will increase substantially, the enterprise value will fall by 10%, and the IRR will only reach 9.09%. The influence of utilization hours and exchange rates for the plant is basically the same. In the scenario of high load and high exchange rate, the value of the enterprise will increase by 10.15% and 6%, respectively, and the IRR will reach about 15%. In the pessimistic scenario, the value of the company is reduced by about 5%, and the IRR is reduced to about 11%.

The impact of carbon tax and environmental protection requirements is similar to that in Indonesia. In the optimistic scenario, the enterprise value and IRR are close to the baseline. In the pessimistic scenario, the fluctuation of the enterprise value for the Vietnam Coal Power Project was larger than that of Indonesia. In addition, the project's IRR fell to 9.49%, in a state of loss, once environmental requirements were taken into account.

Vietnam faces challenges in terms of coal power development: (1) Even in an optimistic scenario, reliance on coal imports by Vietnam will increase the coal price, which will finally decrease the corporate value and IRR. If the coal price reaches the international average, the company will be in a loss state. (2) Coal-fired projects are likely to run into the risk of loss once the Vietnamese government increases the carbon tax and environmental tax and does not provide electricity-price subsidies.

Although the case studies reported in this paper only focused on two countries, the environmental risks studied in this paper are common problems faced by China's overseas coal power investment. For example, all coal power projects will face certain kinds of resource/supply issues. Though most "Belt and Road" countries do not have a carbon tax or environmental taxes to date, the pressure is there to realize the NDC targets. Of course, some factors are case-specific, for example, the exchange rate for Indonesia is due to its unstable financial environment. In addition, although the basic information and results in this paper are only provided for these cases, we argue here that the research framework presented here has universal reference to Chinese companies and government.

## 6. Conclusions and Policy Implications

### 6.1. Conclusions

This paper developed an environmental stress testing tool to examine the environmental risks of China's overseas coal power investment. We elaborated the overall framework of stress testing and provided an in-depth case study to showcase how best to employ the proposed method: Vietnam and Indonesia were used as two national cases and two specific case coal power projects. Our conclusions are as follows:

(1) China needs to incorporate environmental risk into the investment decisions of overseas coal power projects. The environmental stress test method is helpful for coal power enterprises and financial institutions to analyze the impact of environmental risks on the financial situation of enterprises, and to guide environmental risk management.

(2) For Vietnam and Indonesia: Currently, China is investing extensively in these two nations. Our study revealed that the IRR for Chinese coal power investment was 13.95% in Indonesia and 12.35% in Vietnam. As for comparison, a follow-up project-level case study also confirmed the results at the national level. First, affected by the national environment, the economics of coal-fired projects in Indonesia was most sensitive to the exchange rate, while the economics of coal projects in Vietnam was most sensitive to coal price. In the pessimistic scenario, the IRR of the projects could only reach 10% and 9%, respectively.

(3) The enterprise value and IRR of coal power projects in the "Belt and Road" countries are sensitive to environmental requirements. In recent years, a growing market for coal-fired power in the "Belt and Road" countries has enabled the stable income of Chinese power companies. However, faced with the pressure to achieve NDC goals, "Belt and Road" countries will strengthen environmental regulation in the future, which will have a certain impact on the economy of Chinese coal power investment projects. In the case of high environmental taxes and low subsidies, the coal power projects will likely be in a state of loss.

### 6.2. Policy Implications

The results found in this paper have important policy implications, as summarized below:

(1) The power company should be responsible for the investment and operational management of coal-fired power projects in the "Belt and Road" countries. The power company should: (1) evaluate the environmental risks including the coal resource condition, coal power planning, and exchange rate fluctuation before making investment; (2) seek strategies to increase the revenue and decrease costs such as usage of low-cost coal, high technology, increased operating load of the power plant and utilization hours; and take precautions against risks including exchange rate change, carbon tax increase; and, (3) finally, pay attention to the environmental impact, shift more investment to clean energy sector, and promote the green transformation and sustainable development of the world economy.

(2) The financial institutions should strengthen overseas financial services and risk control capabilities, implement strict technical standards for traditional fossil energy projects, and

conduct environmental stress tests. When making financial and investment decisions, they should require power companies to provide environmental information of the projects and factors that may lead to financial risks. At the same time, financial institutions should take environmental considerations into account and invest more in green fields and enterprises. In addition, the current methods and tools of the environmental stress test are not mature, so financial institutions should strengthen cooperation with corporate stakeholders to improve the methods and tools of environmental risk analysis.

(3) The Chinese government should also take several steps to provide a healthy investment environment. On one hand, it should establish a sound legal security system to protect the interests of power companies "going global". On the other hand, it should strengthen diplomatic cooperation with governments of the "Belt and Road" countries in order to create a healthy investment environment. Finally, the Chinese government should establish a coordination mechanism for foreign investment, strengthening guidance for power companies and preventing vicious competition.

(4) The national governments of "the Belt and Road" countries should strictly control the environmental impact of the coal-fired power and strengthen environmental supervision. Faced with the pressure to achieve the NDC goal, on the one hand, governments should gradually improve the pollution emission standards of coal power to build a clean, efficient, and low-carbon power industry system. On the other hand, they should develop renewable energy, introduce a clear renewable energy policy and development plan with clear goals, and gradually improve the localization of renewable energy. In addition, they should provide certain price subsidy incentives to projects with high environmental protection input costs in order to encourage environmental protection solutions in coal power projects.

**Author Contributions:** J.Y. designed the research framework; X.Y. drafted the paper and collected the data for analysis; M.X., S.C. and F.S. polished the manuscript and provided valuable opinions during the revision.

**Funding:** This research was funded by the Fundamental Research Funds for the Central Universities (2018ZD14), the funding of National Natural Science Foundation of China (71673085) and the Beijing Social Science Fund (16YJB027). And the APC was funded by the Energy Foundation.

**Conflicts of Interest:** The authors declare no conflict of interest.

## Appendix A

The calculation of stress test factors in the free cash flow discount method.

Changes in stress test factors will mainly lead to changes in corporate income and cost. In terms of operating income:

$$\text{Main business income}_{(n)} = \text{Priority generation revenue}_{(n)} + \text{Medium and long term market electricity revenue}_{(n)} + \text{Spot market electricity revenue}_{(n)} \tag{A1}$$

$$\text{Priority generation revenue}_{(n)} = \text{Electricity sales} \times \text{Priority generation ratio} \times \text{Coal benchmark feed} - \text{in tariff} \tag{A2}$$

$$\text{Electricity sales} = \text{Utilization hours} \times \text{Installed capacity} \times (1 - \text{Self} - \text{use rate of power plant}) \tag{A3}$$

$$\text{Priority generation ratio} = 100\% - (\text{Medium and long term market power ratio} + \text{Spot market power ratio}) \tag{A4}$$

$$\text{Coal benchmark feed} - \text{in tariff} = \text{Benchmark feed} - \text{in tariff} + \text{Electricity subsidy for desulfurization, denitrification and dust removal} \tag{A5}$$

$$\text{Medium and long term electricity market revenue}_{(n)} = \text{Electricity sales} \times \text{Medium and longterm market power ratio} \times (\text{Coal benchmark feed} - \text{in tariff} - \text{Market electrovalency}) \tag{A6}$$

$$\text{Spot market electricity revenue}_{(n)} = \text{Electricity sales} \times \text{Spot market electricity ratio} \\ \times \text{Spot market price in local currency} \times \text{Exchange rate(RMB against local currency)} \tag{A7}$$

It can be seen from (A1)–(A8) that the utilization hours affect revenue by affecting electricity sales. Considering that the purchase and sale of electricity are mostly paid for in the local currency, this paper focused on the electricity price to show the changes in the exchange rate.

In terms of operating cost:

Changes in coal prices will directly affect fuel costs.

$$\text{Fuel cost }_{(n)} = \text{Power generation coal consumption} \times (1 - \text{Reduction rate of coal consumption })^{n-1} \\ \times \text{Coal price} \times (1 + \text{The rate of increase in coal price})^{n-1} \times \text{Power generation} \tag{A8}$$

Carbon tax and environmental taxes will affect the cost of pollution emissions.

$$\text{Pollution emission cost} = \text{SO}_2 \text{ emission cost} + \text{NO}_x \text{ emission cos t} \\ + \text{Soot emission cost} + \text{CO}_2 \text{ emission cos t} \tag{A9}$$

$$\text{SO}_2 \text{ emission cost} = \text{Power generation} \times \text{SO}_2 \text{ emission factor} \times \text{SO}_2 \text{ environmental tax} \tag{A10}$$

$$\text{NO}_x \text{ emission cost} = \text{Power generation} \times \text{NO}_x \text{ emission factor} \times \text{NO}_x \text{ environmental tax} \tag{A11}$$

$$\text{Soot emission cost} = \text{Power generation} \times \text{Soot emission factor} \times \text{Soot environmental tax} \tag{A12}$$

$$\text{CO}_2 \text{ emission cos t} = \text{Carbon price} \times (1 + \text{Carbon price growth rate}) \times (\text{CO}_2 \text{ emissions} - \text{Available quota}) \\ + (\text{Carbon price} - \text{Carbon auction price}) \times \text{Available quota} \times \text{Carbon market paid quota ratio} \tag{A13}$$

Note: The "*n*" in the above formulas represents the calculation period of year *n*.

## Appendix B

(1) Expected utilization hours

Indonesia's utilization hours are calculated using the generation capacity and installed capacity. Vietnam's data are available online (https://mp.weixin.qq.com/s/VG8OFajftYJ4Q-VgmWHNMg) [62].

$$\text{Utilization hours} = \text{output/average capacity of generating equipment}$$

**Table A1.** Indonesia's expected utilization hours in 2011–2017.

| Year | Unit | 2011 | 2012 | 2013 | 2014 | 2015 | 2016 | 2017 |
|---|---|---|---|---|---|---|---|---|
| Installed coal power capacity | MW | 14,677 | 18,747 | 20,666 | 21,746 | 24,197 | 25,357 | 28,584 |
| Coal power generation | TW·h | 81.09 | 102.17 | 111.25 | 119.53 | 130.51 | 135.4 | 148.3 |
| Utilization hours | h | 5524.9 | 5449.7 | 5383.3 | 5496.7 | 5393.5 | 5339.7 | 5188.2 |

Data sources: IEA database statistics, IRENA, and Coal Plant Tracker.

(2) Coal consumption rate

The coal consumption of power supply refers to the data of typical conventional coal-fired generating units in China.

**Table A2.** Reference value of power supply coal consumption of typical conventional coal-fired generating sets (g/kWh).

| Unit Type | | New Unit | Active Unit | |
| --- | --- | --- | --- | --- |
| | | | Average Level | Advanced Level |
| 1000 MW ultra-supercritical | wet cold | 282 | 290 | 285 |
| | air-cooled | 299 | 317 | 302 |
| 600 MW ultra-supercritical | wet cold | 285 | 298 | 290 |
| | air-cooled | 302 | 315 | 307 |
| 600 MW supercritical | wet cold | 303 | 306 | 297 |
| | air-cooled | 320 | 325 | 317 |
| 600 MW subcritical | wet cold | - | 320 | 315 |
| | air-cooled | - | 337 | 332 |
| 300 MW supercritical | wet cold | 310 | 318 | 313 |
| | air-cooled | 327 | 338 | 335 |
| 300 MW subcritical | wet cold | - | 330 | 320 |
| | air-cooled | - | 347 | 337 |

Data source: National Energy Commission of China Action plan for energy conservation and emission reduction upgrading and upgrading of coal power plants (2014–2020).

(3) Coal price

**Table A3.** Indonesian coal price (dollars/ton).

| Month | 2012 | 2013 | 2014 | 2015 | 2016 | 2017 |
| --- | --- | --- | --- | --- | --- | --- |
| January | 109.29 | 87.55 | 81.90 | 63.84 | 52.00 | 85.00 |
| February | 111.58 | 88.35 | 80.44 | 62.92 | 52.00 | 83.00 |
| March | 112.87 | 90.09 | 77.01 | 67.76 | 52.00 | 82.50 |
| April | 105.61 | 88.56 | 74.81 | 64.48 | 52.00 | 84.00 |
| May | 102.12 | 85.33 | 73.60 | 61.08 | 52.00 | 84.00 |
| June | 96.65 | 84.87 | 73.64 | 59.59 | 53.00 | 75.46 |
| July | 87.56 | 81.69 | 72.45 | 59.16 | 54.00 | 80.00 |
| August | 84.65 | 76.70 | 70.29 | 59.14 | 59.14 | 84.65 |
| September | 86.21 | 76.89 | 69.69 | 58.21 | 61.00 | 88.00 |
| October | 86.04 | 76.61 | 67.26 | 57.39 | 67.30 | 89.00 |
| November | 81.44 | 78.13 | 65.70 | 54.43 | 81.44 | 91.00 |
| December | 81.75 | 80.31 | 69.23 | 53.51 | 101.69 | 90.00 |
| Average | 95.48 | 82.92 | 73.00 | 60.13 | 61.46 | 84.72 |
| Convert to RMB (yuan/ton) | 658.95 | 572.01 | 500.94 | 414.69 | 424.07 | 584.57 |

Data source: Indonesian Coal Mining Association (APBI).

Vietnam's coal prices are based on a report (Carbon Tracker, Economic and financial risks of coal power in Vietnam, 2018).

(4) Unit kilowatt investment cost

Investment cost of generation = total investment (yuan)/total installed coal power (kW)

The figures are based on existing coal-fired power plants in these two countries.

**Table A4.** Part of Indonesia's coal power project unit kilowatt investment cost.

| Project | Installed Capacity | Type | Location | Unit Kilowatt Investment Cost (Yuan/kW) |
|---|---|---|---|---|
| Java-9&10 | 2000 | Coal-fired | Banten | 10,647 |
| Meulaboh-3&4 | 400 | Coal-fired | Aceh | 8505 |
| Sumut-2 | 600 | Coal-fired | North Sumatera | 9450 |
| Kalbar-2 | 200 | Coal-fired | West Kalimantan | 9450 |
| Sumbagsel-1 | 300 | Coal-fired | Southern Sumatera | 9450 |
| Sulbagut-3 | 100 | Coal-fired | Northern Sulawesi | 9450 |
| Jambi-I | 600 | CMM | Jambi | 9450 |
| Kaltim-5 | 200 | CMM | East Kalimantan | 9450 |
| Kalselteng-3 | 200 | CMM | South/Central Kalimantan | 9450 |
| Kaltim-3 | 200 | CMM | East Kalimantan | 9450 |
| Kaltim-6 | 200 | CMM | East Kalimantan | 9450 |
| Sumsel-6 | 600 | CMM | South Sumatera | 9450 |
| Riau-1 | 600 | CMM | Riau | 9450 |
| Jambi-II | 600 | CMM | Jambi | 9450 |
| Kalselteng-4 | 200 | CMM | South/Central Kalimantan | 9450 |
| Kalselteng-5 | 200 | CMM | South/Central Kalimantan | 9450 |
| Jawa-7 | 2000 | Coal-fired | Banten | 5935 |

Data source: PwC Power in Indonesia Investment and Taxation Guide, Sina Finance News.

**Table A5.** Part of Vietnam's coal power project unit kilowatt investment cost.

| Project | Installed Capacity | Investment Cost (Billion Dollars) | Unit Kilowatt Investment Cost (Yuan/kW) |
|---|---|---|---|
| Vietnam Vinhtan power plant | 2*600 MW | 1.755 | 10,091.25 |
| Nam dinh coal power plant | 2*600 MW | 2.2 | 12,650 |
| Vietnam's Haiyang coal power plant | 2*600 MW | 1.8 | 10,350 |
| Average | | | 11,030.42 |

Data Sources: Polaris power network. Available online: Vietnam Vinhtan power plant (http://news.bjx.com.cn/html/20180504/895578.shtml) [63]; Nam dinh coal power plant (http://news.bjx.com.cn/html/20180518/898654.shtml) [64]; Vietnam's Haiyang coal power plant (http://news.bjx.com.cn/html/20180719/914089.shtml) [65]

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
