# Peer review of "Environmental Stress Testing for China’s Overseas Coal Power Investment Project"

_sustainability, doi:10.3390/su11195506_

Round 1
Reviewer 1 Report
The research design is flawed in general. This article does not provide a meaningful research question. I don't really understand how the literature is related to what authors try to address. They also need to highlight their contributions to this paper rather than applying an existing methodology to reach a common-sense conclusion.
There are conceptual problems in presenting the findings. in Page 17, figure 12 shows four indicators-Pessimistic, Optimistic-Pessimistic, Average and GH EMM. what do those concepts mean?
For the policy implications, authors should expand their audience instead of narrowing down to the Chinese government and Chinese companies. How the host country should respond if they wish to attract foreign investment to help investors mitigate the risks?
Author Response
Dear reviewer:
We are very grateful to your comments for the manuscript. All of these comments have contributed a lot to improve the quality of our article. According to your advice, we amended the relevant part in manuscript. Revised manuscript is marked by using the "Track Changes" function in the paper, so that changes are clearly highlighted and easily visible. For clarity, below is a point-by-point response to the comments.
Point 1: The research design is flawed in general. This article does not provide a meaningful research question. I don't really understand how the literature is related to what authors try to address. They also need to highlight their contributions to this paper rather than applying an existing methodology to reach a common-sense conclusion.
Response 1: Thank you for your kind comment. This paper studies the impact of environmental risks in "Belt & Road" countries on the economy of China's overseas coal power investment projects. To clarify the research significance of this question, we added following research background in Section 1. From Chinese side, the power sector’s enthusiasm in oversea coal power investment is largely driven by economic concern: coal power projects are capital intensive investment and its return is stable and substantial. This is especially attractive for Chinese coal power utilities who are accustomed with the stable return expectations within a highly planned power system regulation under strong market growth. However, with the recent advent of new economical normal, power demand growth slows down and renewable energy grows quickly in China and leads to overcapacity in coal power and a radical structural change in the power market. As a result, power utilities switch to overseas investment. But from the international perspective, China’s active involvement in BRI coal power investment represents a big challenge to global efforts stabilizing greenhouse gases emissions, given the fact that industrialized economies have reached a consensus on divestment in coal. Though Chinese economic consideration and the climate change concern of the international society are not easily reconciled, a deep dive into Chinese economic consideration can shed more insight into the emissions abatement vision. As a matter of fact, what happened to coal (divestment) in industrialized economies and what the impact of structural change to coal power will be in China will certainly happen in BRI developing economies sometimes later. Therefore, even from Chinese perspective, a more considerate strategy incorporating the potential risks of structural & market regulation changes in these BRI host countries and the long-term environmental & climate change risks into decision-making process can possibly reconcile the pervasive contradiction. In addition, we illustrated the contribution of this article in the last paragraph of Section 2. First, this paper identifies the risk factors related with China’s overseas coal power investment and present a stress testing framework. Second, this paper examines the sensitivity of coal project value in two case countries from the perspective of environmental risks.
Point 2: There are conceptual problems in presenting the findings. in Page 17, figure 12 shows four indicators-Pessimistic, Optimistic-Pessimistic, Average and GH EMM. what do those concepts mean?
Response 2: Thanks so much for your comment. Pessimistic refers to the outcome of pessimistic situation. Optimistic-Pessimistic refers to the difference between the optimistic scenario and the pessimistic scenario. The average refers to the average level of Indonesia.GH EMM refers to the GH EMM Indonesia Project. And we have added this in the Section 5.
Point 3: For the policy implications, authors should expand their audience instead of narrowing down to the Chinese government and Chinese companies. How the host country should respond if they wish to attract foreign investment to help investors mitigate the risks?
Response 3: Thanks for your kind comment. In this version, we have made the following revisions in Section 6 according to your insightful comments: Firstly, in the Section 6.1, we extended the impact of environmental risks on China's coal-power investment projects to the whole "Belt & Road" countries for analysis, rather than just the two countries studied in this paper. Secondly, we expanded the policy implications in Section 6.2, analysed the relevant policy suggestions in detail, and added the policy suggestions for the government of "Belt & Road" countries.
Thank you again for your advice and hope to learn more from you. If you have any question about this paper, please don’t hesitate to let us know.
Kind regards,
Authors

Reviewer 2 Report
In this paper, the authors address an important and topical issue: environmental and energy concerns of China's Belt and Road Initiative. The potential significance of this study lies in the fact that the BRI is development of global scale, and will have implications for a growing number of countries and in many different dimensions over the coming years and decades. Understanding how the BRI interfaces with environmental and energy concerns is of great importance and urgency. However, these aspects of the BRI also make it challenging for a study to address it in a satisfactory way. Most significantly, because the BRI is so fast moving and diverse, a study of one particular dimension of it and focused on a small subset of countries - as this study does - can be easily overtaken by events and be quickly made irrelevant. To that end, I suggest two major revisions that must be made to this paper if it is to have lasting relevance as a peer-reviewed academic publication.
First, the authors should provide a more thorough justification for their chosen methodology - e.g., better explain why the testing factors they have chosen are good indicators by better contextualizing them in other, similar studies.
Second, the authors need to greatly expand on the Conclusion and Policy Implications section. After a substantial amount of methods and results, this section is way too skimpy. Indeed, this section is in many ways the most important section of the paper, as it gives the authors an opportunity to make a general case for why their results are significant and representative of BRI at large. This section needs to more closely resemble a substantial Discussion section, with many references of relevant studies and also charting a clear path forward for future research. As it stands, this section does not come close to approximating that level of exposition.
Finally, as a minor comment - the language flows relatively smoothly but is still hobbled occasionally by grammatical errors and stylistic awkwardness. It could benefit from a more thorough read-through and language editing.
Author Response
Dear reviewer:
We are very grateful to your comments for the manuscript. All of these comments have contributed a lot to improve the quality of our article. According to your advice, we amended the relevant part in manuscript. Revised manuscript is marked by using the "Track Changes" function in the paper, so that changes are clearly highlighted and easily visible. For clarity, below is a point-by-point response to the comments.
Point 1: First, the authors should provide a more thorough justification for their chosen methodology - e.g., better explain why the testing factors they have chosen are good indicators by better contextualizing them in other, similar studies.
Response 1: Thank you for your kind suggestion. According to your comment, we have made corresponding modifications in the Section 3.2. Firstly, we elaborated on the major parties involved in Chinese overseas coal-power projects and their agreement relations through Figure 2. Secondly, we analysed the environmental risks in construction, operation and financing phases of coal power projects, and describe the reasons why this paper chooses to study the risks of coal resources, coal power planning, exchange rate, carbon tax and environmental protection.
Point 2: Second, the authors need to greatly expand on the Conclusion and Policy Implications section. After a substantial amount of methods and results, this section is way too skimpy. Indeed, this section is in many ways the most important section of the paper, as it gives the authors an opportunity to make a general case for why their results are significant and representative of BRI at large. This section needs to more closely resemble a substantial Discussion section, with many references of relevant studies and also charting a clear path forward for future research. As it stands, this section does not come close to approximating that level of exposition.
Response 2: Thanks so much for your comment. In this version, we have made the following revisions in Section 6 according to your insightful comments: Firstly, in the Section 6.1, we extended the impact of environmental risks on China's coal-power investment projects to the whole "Belt & Road" countries for analysis, rather than just the two countries studied in this paper. Secondly, we expanded the policy implications in Section 6.2, analysed the relevant policy suggestions in detail, and added the policy suggestions for the government of "Belt & Road" countries.
Point 3: Finally, as a minor comment - the language flows relatively smoothly but is still hobbled occasionally by grammatical errors and stylistic awkwardness. It could benefit from a more thorough read-through and language editing.
Response 3: Thanks for your kind comment. According to your suggestion, we have made read-through and revised the language in the revised manuscript.
Thank you again for your advice and hope to learn more from you. If you have any question about this paper, please don’t hesitate to let us know.
Kind regards,
Authors

Reviewer 3 Report
This is a good idea for a paper.
It is a worthwhile exercise.
Some high level comments:
Is this about overseas investment or about coal per se? This would look similar in China itself. Need to emphasise the risk that is different from in China itself. It would be a good idea to have a benchmark Chinese project in the analysis to show this… Why have two countries – are they sufficiently different? The comparison table does not show much different between them – need to bring out substantial differences between them (or just concentrate on one country). Or compare one of them to a project in China. The stress testing should combine the risks, a Monte Carlo risk analysis would be much better. The overall results don't stress the projects enough – as we have seen in Europe coal plants have been driven out of business by high carbon prices (up to 50 USD/tonne in the UK) and the requirement to have FGD installed to deal with SO2.
Detailed comments:
Line 30: this is to do with promoting exports.
Line 36: 1.1%? this is low?
Lines 42-44: This is a testable proposition and government statement, who says this?
Line 213: hook relationship = ?
Line 299: are shelved and cancelled projects significant? – don't include them if they are.
Line 349: coal prices could fall, or be hedged via electricity market price adjustments?
Line 381: price could go down…has due to Shale gas.
Line 399: rose above?
Figure 8: Show over 10 years. Further risk if Yuan appreciates relative to the dollar.
Line 458: Statistical Review of World Energy
Line 491: Show same depreciation in exchange rate or explain the depreciation over 10 years has been higher in one or the other…
Line 557: the ranking is an arbitrary function of the degree to which you stress each of the variables.
Author Response
Dear reviewer:
We are very grateful to your comments for the manuscript. All of these comments have contributed a lot to improve the quality of our article. According to your advice, we amended the relevant part in manuscript. Revised manuscript is marked by using the "Track Changes" function in the paper, so that changes are clearly highlighted and easily visible. For clarity, below is a point-by-point response to the comments.
Point 1: Is this about overseas investment or about coal per se? This would look similar in China itself. Need to emphasise the risk that is different from in China itself. It would be a good idea to have a benchmark Chinese project in the analysis to show this… Why have two countries – are they sufficiently different? The comparison table does not show much different between them – need to bring out substantial differences between them (or just concentrate on one country). Or compare one of them to a project in China.
Response 1: Thank you for your comment. Firstly, this paper focuses on the impact of the environmental risks faced by Chinese coal power investment in “Belt & Road” countries on the economics of coal power projects. Under China's current market structure, power companies believe that investing in coal power projects can achieve stable expected returns. Similarly, when investing overseas, it is assumed that coal power projects in developing countries can get a stable expected return. Secondly, Indonesia and Vietnam are chosen as case nations because they are the largest coal power FDI destination countries of China. Though our purpose is not for comparative study, the difference in environmental factors of these two countries can reveal the sensitivity of stress testing due to difference in national status, market and regulations. And we have added this in Paragraph 5,Section 1.
Point 2: The stress testing should combine the risks, a Monte Carlo risk analysis would be much better.
Response 2: Thank you for your kind suggestion. As a statistical simulation method, Monte Carlo is based on the statistical sampling theory. It uses random numbers to conduct sampling experiments or random simulations on the existing data of random variables, so as to study the influence of risk factors. Since there are not enough data sources in this paper, it is difficult to analyse the risk variation through this method. However, the data in the optimistic and pessimistic scenarios set in this paper have certain basis, so we believe that the result is reasonable.
Point 3: The overall results don't stress the projects enough – as we have seen in Europe coal plants have been driven out of business by high carbon prices (up to 50 USD/tonne in the UK) and the requirement to have FGD installed to deal with SO2.
Response 3: Thank you for your comment. For the scenario setting of carbon price and environmental taxes, considering that the carbon market in "Belt & Road" countries has just started, our consideration of carbon price and environmental protection tax is relatively low. For example, the cost of power generation in China has increased by nearly 10% from the relatively low emission standard to the current ultra-low emission standard. But a direct assumption of 10% growth is not particularly realistic in the near to medium term for “Belt & Road” countries. Even so, if the government don’t provide the subsidies for Chinese power enterprises, the coal power projects will also suffer steep declines in efficiency. And we have added the corresponding statement in the last paragraph of Section 4.2.
Point 4: Line 30: this is to do with promoting exports.
Response 4: Thank you for your comment. In our view, import and export trade and connectivity are channels and ways to promote common prosperity and development, rather than the ultimate goal of “Belt & Road” initiative. And we modified the statement in the revised manuscript.
Point 5: Line 36: 1.1%? this is low?
Response 5: Thank you for your comment. The 1.1% refers to the annual growth rate of China's total trade with "Belt & Road" countries. For the coal power investment sector, the figure will much higher.
Point 6: Lines 42-44: This is a testable proposition and government statement, who says this?
Response 6: Thank you for your comment. This is put forward in “Vision and Action for Promoting Energy Cooperation between the Silk Road Economic Belt and the 21st Century Maritime Silk Road” published by the national energy administration, and we have added the relevant reference in the revised manuscript.
Point 7: Line 213: hook relationship = ?
Response 7: Thank you for your comment. “Hook relationship” refers to the relationship between the relevant figures in financial statements, which can be used for mutual examination and verification. And we added this in the revised manuscript.
Point 8: Line 299: are shelved and cancelled projects significant? – don't include them if they are.
Response 8: Thank you for your comment. The proportion of shelved or cancelled projects is not high. But we take them into account because we believe they also reflect Chinese power companies' willingness to invest in coal power projects overseas.
Point 9: Line 349: coal prices could fall, or be hedged via electricity market price adjustments?
Response 9: Thanks so much for your comment. As we mentioned in Section 4.1, according to Vietnam's current coal source for power generation, 75% of the coal depends on import and will be subject to volatile in international coal market. We believe that the government from the perspective of protecting the competitiveness of electricity price, will not shift the risk of coal cost fluctuation to the downstream indefinitely. Therefore, we believe that Vietnam's coal prices will rise due to the volatility of the international coal market.
Point 10: Line 381: price could go down…has due to Shale gas.
Response 10: Thank you for your comment. We quite agree with you, it is true that shale gas revolution in the USA has lead to a radical rebalance of primary energy market. If shale gas is taken into account, coal price are likely to go down. But the long-term impact of shale gas revolution, and its impact to Asia-pacific energy market is beyond the scope of this study. If possible, we will consider this in our next study.
Point 11: Line 399: rose above?
Response 11: Thank you for your kind comment. We have modified the “fell below” to “rose above” in the revised manuscript.
Point 12: Figure 8: Show over 10 years. Further risk if Yuan appreciates relative to the dollar.
Response 12: Thank you very much for your advice. According to your suggestions, we have made the following modifications in the revised manuscript. Firstly, in the exchange rate of Section 4.1, in order to reflect the risk of Chinese Yuan(CNY) appreciates relative to the dollar, we added the chart of CNY relative to the Indonesian rupiah(IDR) and Vietnamese dong(VND). The data of CNY relative to IDR from 2001 to 2019 were selected, and the data of CNY relative to VND from 2008 to 2019 were presented. Secondly, in the Section 4.2, the exchange rate of dollar to IDR(VND) was replaced by the exchange rate of CNY to IDR (VND) in the analysis of the exchange rate risk, and it was synchronously modified in Section 5.
Point 13: Line 458: Statistical Review of World Energy
Response 13: Thank you for your comment. We have modified this statement in the revised manuscript.
Point 14: Line 491: Show same depreciation in exchange rate or explain the depreciation over 10 years has been higher in one or the other…
Response 14: Thanks for your comment. As mentioned in Response 12, we have provided the exchange rate data of CNY relative to IDR from 2001 to 2019 and the data of CNY relative to VND from 2008 to 2019 in Section 4.1. And in the Table 5 of Section 4.2, we set the optimistic and pessimistic exchange rate data respectively according to the exchange rate fluctuations in the past five years.
Point 15: Line 557: the ranking is an arbitrary function of the degree to which you stress each of the variables.
Response 15: Thank you for your comment, we are sorry for our carelessness. We have modified the “rank” to “order” in the revised manuscript.
Thank you again for your advice and hope to learn more from you. If you have any question about this paper, please don’t hesitate to let us know.
Kind regards,
Authors

Round 2
Reviewer 1 Report
I have the opportunity to review your paper four times in four versions. This one is much better in presenting your research design that concerns me the most.
Reviewer 2 Report
The authors have made substantial changes to the manuscript, which have made the study more intelligible and correspondingly improved the quality of the paper. However, it still needs to be thoroughly edited for language - for instance, in the very first sentence of the Abstract: "The advance of Chinese “Belt and Road” Initiative encourages increased overseas investment in coal power projects." This should be: "The advance of China's “Belt and Road” Initiative encourages increased overseas investment in coal power projects." Other such errors are littered throughout the manuscript and need to be corrected before acceptance and publication.
This manuscript is a resubmission of an earlier submission. The following is a list of the peer review reports and author responses from that submission.
Round 1
Reviewer 1 Report
Review comments
In general, I am not sure how meaningful the research question is or if there is a research question. The author just basically list the literature without clearly illustrate how their paper could contribute to the intellectual debate. The whole piece is rather descriptive than analytical. The authors need to work on many aspects in order to make it a do-able research paper.
1. have a meaningful research question
2. explain the reasons for case selections: why Vietnam and Indonesia.
2. how their research is applicable to other Chinese projects since they only focus on two projects in Vietnam and Indonesia.
3. explain what purposes they want to achieve in this paper. I get lost when I read the paper.
4. the policy suggestions and conclusions seem to be irrelevant from their modelling. Without the modelling, people can still make similar conclusions. So be specific why they think their modelling is helpful in explain what research question.
Some minor errors,
In page 3, line 2, “ the author of (17) conducted …” what is the 17? The same applies to 18 . also, in (26), (32) and (33), ect. Please double check your paper before submission. Those are incomplete sentences.
In summary, this paper is far from being complete. In forms, there are errors and missing information. In content, it just does not meet the requirement of a research paper.
Author Response
Dear reviewer:
I am very grateful to your comments for the manuscript. All of these comments have contributed a lot to improve the quality of our article. According to your advice, we amended the relevant part in manuscript. Some of your questions were answered below.
Point 1: The author just basically list the literature without clearly illustrate how their paper could contribute to the intellectual debate. The whole piece is rather descriptive than analytical.
Response 1: This paper introduces the previous literatures from three aspects(stress test method, overseas power investment, power development in Indonesia and Vietnam). Based on your comment, we made the following modifications in the Section 2:
(1) In terms of cited literatures, we reduced literatures related to stress test in the financial sector, added some literatures related to power investment risks, and adjusted the order of some literatures.
(2) We added the relevant summaries of previous literatures in each part. The first part points out the feasibility of the stress test method, the second part points out that the impact of environmental risks on overseas coal power projects is in a blank state, and the third part points out that Indonesia and Vietnam have a huge coal power development market.
(3) We added the relevant contributions of this paper in the last paragraph.
Point 2: have a meaningful research question.
Response 2: This paper studies the impact of environmental risks in "Belt & Road" countries on the economy of China's overseas coal power investment projects. To clarify the research significance of this question, we added relevant research background into the revised manuscript.
(1) In the third paragraph of the Section 1, we introduced the international controversy faced by China in the overseas investment of coal power.
(2) In the fourth paragraph of the Section 1, we introduced the environmental risks faced by overseas coal power projects that have been put into operation or will be put into operation.
Point 3: explain the reasons for case selections: why Vietnam and Indonesia.
Response 3: There are two main reasons for choosing Indonesia and Vietnam as cases.
Firstly, in recent years, China's coal power investment in these two countries is relatively large among "Belt & Road" countries. The study of coal power projects in these two countries are representative. In the second paragraph of the Section 1, we have made relevant background elaboration and provided relevant data proof in Appendix A.
Secondly, Indonesia and Vietnam are facing many environmental risks, which is of great research significance. In the fourth paragraph of the Section 1, we gave a brief introduction to the environmental risks of the two countries.
Point 4: how their research is applicable to other Chinese projects since they only focus on two projects in Vietnam and Indonesia.
Response 4: In the last paragraph of Section 5, we added the reference of this paper to other coal power projects. Which indicates that the environmental risks studied in this paper are mostly common problems faced by "Belt & Road" countries, and some risks are specifically studied for the case projects.
Point 5: explain what purposes they want to achieve in this paper. I get lost when I read the paper.
Response 5: The research of this article has two main purposes. On the one hand, we hope to prove that Chinese power companies need to consider relevant environmental risks when investing in coal power projects overseas. On the other hand, we hope to analyse the sensitivity of coal power project value in two case countries from the perspective of environmental risks. The revised details can refer to the Response 1.
Point 6: the policy suggestions and conclusions seem to be irrelevant from their modelling. Without the modelling, people can still make similar conclusions. So be specific why they think their modelling is helpful in explain what research question.
Response 6: Based on your comment, we made same revisions in the Page 17,Section 6.
In the Section 6.1, we deleted the content that was repeated with the previous result analysis, and added some conclusions related to this paper. In the Section 6.2, we deleted the policy suggestions that were not related to this paper.
Point 7:In page 3, line 2, “ the author of (17) conducted …” what is the 17? The same applies to 18 . also, in (26), (32) and (33), ect. Please double check your paper before submission. Those are incomplete sentences.
Response 7: For these incomplete sentences, we have revised all of them in the Section 2.
Thank you again for your advice and hope to learn more from you. If you have any question about this paper, please don’t hesitate to let us know.
Kind regards,
Authors

Reviewer 2 Report
This paper uses the environmental stress testing to examine the BRI investment risks with regards to environmental issues in the related countries. The case study looks interesting, and also provides useful information for policy-makers and industrial stakeholders. Nevertheless, there are some issues in the manuscript regarding content organizing, formatting, wording etc. I would recommend revisions before accepting it for publication. Specific comments are listed below:
General problems:
· In the introduction section, it would be good to give some brief information of the ‘Belt & Road’ countries, e.g. how many countries are involved and what parts of the world regions are covered? why did you choose Indonesia and Vietnam as the two cases?
· The literature review needs to be improved. The review on regular stress testing in the financial sector is unnecessary, as this is a quite common practice. Please refine this part and focus on environmental risk analysis and environmental stress testing, which are not sufficiently discussed in this section.
· Please provide the spell-out versions for many abbreviations when used for the first time, such as ICBC, IRR, EBIT, WACC, IPP, PPA, PLN, EPC, IRENA, BAU, NDC etc.
· In the case study, some factors, such as coal price, exchange rate and utilization hours, seem not in the scope of ‘environmental risk factors’, as they have no close relation with environmental issues. Perhaps it would be good to add some explanation somewhere that this is an ‘expanded’ environmental stress testing.
Specific problems
· Page 1, Section 1, Introduction, “The “Belt & Road” countries generally have lower energy utilization technologies and power industry level compared to other more countries” --- this sentence is not clear, what is ‘lower energy utilization technologies? compared to what countries’ ? please clarify
· Page 2, Section 1, ‘These countries cover a total population of 4.6 billion, but the per capita electricity consumption is about 2,825 kWh, which is far below the international level of 3,295 kWh’ – the data for which year?
· Page 2, Section 1, ‘In 2015, coal-fired power installations in these countries reached 1.398 billion kWh’ – the unit looks weird, the installations should be measured by power unit, such as MW, GW, TW, kWh is energy unit, typically, it is more common to use MWh, GWh or TWh for higher order scales. I would suggest the authors unify the units throughout the manuscript, as there are quite some weird units shown in the paper.
· Page 4, Section 3.1, how is Eq. (4) related to Eqs. (1) – (3)? the variables in Eq. (4) seem irrelevant from other equations, please clarify.
· Page 5, Section 3.2. Environmental Risk Analysis. ‘Most IPP projects are funded by financing.’ – financing is a very general concept, what kinds of financing channels in this case?
· Page 6, Section 4.1 The equivalent available coefficient was 98.75%, and the market share reached 119%’ – what is the equivalent available coefficient? Do you mean the availability factor? what is the market share?
· Page 6, Section 4.1 ‘In 2014, It continued to be the most reliable, nonstop power plant in the South Jiangsu region, with an annual power generation capacity of 2.033 billion kWh’ – should it be South Sumatra? Still, the unit looks weird
· Page 6, Section 4.1 ‘Ranked top among all operating power plants in China’ – should it be in Indonesia?
· Page 7, Section 4.1 ‘Sulfur, nitrogen and soot emissions were 0.35g/kWh, 0.74g/kWh and 0.04 g / kWh, far above Indonesia's emissions standards’ – far below the standards?
· Page 7, Section 4.2. ‘In 2017, Indonesia's coal output was 407 million tons, and domestic coal consumption was 8,580 tons’ – only 8 thousand tons? please add data source
· Page 8, Section 4.2, Figure 3 the unit of y axis is also weird, do you mean tonne of coal equivalent (TCE)
· Page 10, Table 1, title, what is the meaning of ‘coal-fired countries’ ?
· Page 11, ‘Vietnam's NDC goal is to reduce the carbon intensity of GDP by 20% in 2013’ -- reduce the intensity by 20% in 2030?
· Page 13, Table 3, for the Vietnam case, it is quite strange that the coal price variation in the optimistic and pessimistic scenarios is so small (450 and 460 yuan/ton respectively ), is it true that you state in Page 15 ‘In pessimistic scenario, coal prices will increase substantially’ ? would be good to also show the reference settings in this table
Author Response
Dear reviewer:
We are very grateful to your comments for the manuscript. All of these comments have contributed a lot to improve the quality of our article. According to your advice, we amended the relevant part in manuscript. Some of your questions were answered below.
Point 1: In the introduction section, it would be good to give some brief information of the ‘Belt & Road’ countries, e.g. how many countries are involved and what parts of the world regions are covered?
Response 1: Thank you for your kind suggestion. In the first paragraph of the Section 1, we have added the latest information of the "Belt & Road" countries.
Point 2: why did you choose Indonesia and Vietnam as the two cases?
Response 2: There are two main reasons for choosing Indonesia and Vietnam as cases.
Firstly, in recent years, China's coal power investment in these two countries is relatively large among "Belt & Road" countries. The study of coal power projects in these two countries are representative. In the second paragraph of the Section 1, we have made relevant background elaboration and provided relevant data proof in Appendix A.
Secondly, Indonesia and Vietnam are facing many environmental risks, which is of great research significance. In the fourth paragraph of the Section 1, we gave a brief introduction to the environmental risks of the two countries.
Point 3: The literature review needs to be improved. The review on regular stress testing in the financial sector is unnecessary, as this is a quite common practice. Please refine this part and focus on environmental risk analysis and environmental stress testing, which are not sufficiently discussed in this section.
Response 3: This paper introduces the previous literatures from three aspects(stress test method, overseas power investment, power development in Indonesia and Vietnam). Based on your comment, we made the following modifications in the Section 2:
(1) In terms of cited literatures, we reduced literatures related to stress test in the financial sector, added some literatures related to power investment risks, and adjusted the order of some literatures.
(2) We added the relevant summaries of previous literatures in each part. The first part points out the feasibility of the stress test method, the second part points out that the impact of environmental risks on overseas coal power projects is in a blank state, and the third part points out that Indonesia and Vietnam have a huge coal power development market.
(3) We added the relevant contributions of this paper in the last paragraph.
Point 4: Please provide the spell-out versions for many abbreviations when used for the first time, such as ICBC, IRR, EBIT, WACC, IPP, PPA, PLN, EPC, IRENA, BAU, NDC etc.
Response 4: Thank you for your reminding. The spell-out versions have been added into the revised manuscript.
Point 5: In the case study, some factors, such as coal price, exchange rate and utilization hours, seem not in the scope of ‘environmental risk factors’, as they have no close relation with environmental issues. Perhaps it would be good to add some explanation somewhere that this is an ‘expanded’ environmental stress testing.
Response 5: According to your suggestion, we made the following modifications.
In the fourth paragraph of Section 1 and the first paragraph of Section 3.2, it is pointed out that the environmental risks studied in this paper include not only the natural environment, but also the political environment and economic environment.
Point 6: Page 1, Section 1, Introduction, “The “Belt & Road” countries generally have lower energy utilization technologies and power industry level compared to other more countries” --- this sentence is not clear, what is ‘lower energy utilization technologies? compared to what countries’ ? please clarify
Response 6: The expression of this sentence is problematic, for which we have made the adjustment. The countries compared here refer to the other countries in the world.
Point 7: Page 2, Section 1, ‘These countries cover a total population of 4.6 billion, but the per capita electricity consumption is about 2,825 kWh, which is far below the international level of 3,295 kWh’ – the data for which year?
Response 7: These data are for 2015, which we have added in the revised manuscript.
Point 8: Page 2, Section 1, ‘In 2015, coal-fired power installations in these countries reached 1.398 billion kWh’ – the unit looks weird, the installations should be measured by power unit, such as MW, GW, TW, kWh is energy unit, typically, it is more common to use MWh, GWh or TWh for higher order scales. I would suggest the authors unify the units throughout the manuscript, as there are quite some weird units shown in the paper.
Response 8: Thanks for your careful checks. We are sorry for our carelessness. And we have modified all expressions like "billion kWh" into normative units.
Point 9: Page 4, Section 3.1, how is Eq. (4) related to Eqs. (1) – (3)? the variables in Eq. (4) seem irrelevant from other equations, please clarify.
Response 9: The FCFF in formula 4 is the corporate cash flow calculated in formula 1, and we have added relevant explanations in formula 1.
Point 10: Page 5, Section 3.2. Environmental Risk Analysis. ‘Most IPP projects are funded by financing.’ – financing is a very general concept, what kinds of financing channels in this case?
Response 10: Thanks for your kind advice. In the second paragraph of Section 3.2, we added several major financing channels in China.
Point 11: Page 6, Section 4.1 The equivalent available coefficient was 98.75%, and the market share reached 119%’ – what is the equivalent available coefficient? Do you mean the availability factor? what is the market share?
Response 11: The equivalent availability coefficient refers to the ratio of the available hours of a unit minus the equivalent shutdown hours of the reduced output of the unit and the hours during the statistical period of the unit. The market share refers to the ratio of the sales volume of the company to the sales volume of the largest competitor in the market. Relevant explanations have been added in the revised manuscript.
Point 12: Page 6, Section 4.1 ‘In 2014, It continued to be the most reliable, nonstop power plant in the South Jiangsu region, with an annual power generation capacity of 2.033 billion kWh’ – should it be South Sumatra? Still, the unit looks weird
Response 12: Thank you for your careful revise. It should be the South Sumatra region here. We have modified this in the revised manuscript. And the unit has also been modified.
Point 13: Page 6, Section 4.1 ‘Ranked top among all operating power plants in China’ – should it be in Indonesia?
Response 13: It should be ‘Ranked top among all operating power plants of Guohua Group’ here, which has been modified in the revised manuscript.
Point 14: Page 7, Section 4.1 ‘Sulfur, nitrogen and soot emissions were 0.35g/kWh, 0.74g/kWh and 0.04 g / kWh, far above Indonesia's emissions standards’ – far below the standards?
Response 14: We are sorry for our carelessness. It should be much higher than the standard, which has been modified in the revised manuscript.
Point 15: Page 7, Section 4.2. ‘In 2017, Indonesia's coal output was 407 million tons, and domestic coal consumption was 8,580 tons’ – only 8 thousand tons? please add data source
Response 15: We are sorry for our carelessness. It should be 85.8 million tons, which has been modified. The data from the BP Statistical Review of World Energy 2018.
Point 16: Page 8, Section 4.2, Figure 3 the unit of y axis is also weird, do you mean tonne of coal equivalent (TCE)
Response 16: The unit here should be Million tons of coal equivalent (Mtce), which has been modified in the revised manuscript.
Point 17: Page 10, Table 1, title, what is the meaning of ‘coal-fired countries’ ?
Response 17: Here refers to the major coal power countries, which has been modified in the revised manuscript.
Point 18: Page 11, ‘Vietnam's NDC goal is to reduce the carbon intensity of GDP by 20% in 2013’ -- reduce the intensity by 20% in 2030?
Response 18: There's an error in the statement. It is supposed to reduce the carbon intensity of GDP by 20% in 2030. We have modified this in the revised manuscript.
Point 19: Page 13, Table 3, for the Vietnam case, it is quite strange that the coal price variation in the optimistic and pessimistic scenarios is so small (450 and 460 yuan/ton respectively), is it true that you state in Page 15 ‘In pessimistic scenario, coal prices will increase substantially’ ? would be good to also show the reference settings in this table.
Response 19: Thank you very much for your advice. In the last paragraph of Section 4.3, we have made modifications to the coal price setting under the pessimistic scenario of Vietnam. Considering that Vietnam will be heavily dependent on coal imports in the future, the price of coal was set as the international average price. This paper selects the price of steam coal with a calorific value of 5500.And the corresponding stress test results are modified in Section 5.
Thank you again for your advice and hope to learn more from you. If you have any question about this paper, please don’t hesitate to let us know.
Kind regards,
Authors

Reviewer 3 Report
This study suffers from both minor weaknesses and major flaws. The minor weaknesses have to do with language issues, citation, and occasional incoherence. To begin, the paper contains a good number of grammatical errors that make it difficult to follow at times. Next, it cites a lot of data without providing sources of the pertinent information. This raises questions about the rigor and accuracy of the study. Further, coherence is also a problem. While section 3.1 of the study discusses the free cash flow method, the paper fails to show how the method is actually used in establishing the parameters incorporated into the subsequent environmental stress tests. Similarly, the concluding section includes some recommendations that have little to do with the study. Moreover, when discussing the GH EMM Power Plant in Indonesia on page 6, the authors veered to a discussion of South Jiangsu region, which is abrupt and perplexing.
In addition, this study also contains two major flaws. First, while the literature review provides a long list of relevant studies, it fails to show the gap, debate, or inconsistencies of the literature regarding environmental stress testing in China. Thus, it is thus unclear how this study contributes to the literature. Second, and more importantly, while constructing an optimistic and pessimistic scenario when conducting the environmental stress tests on the two select coal-fired power plants in Indonesia and Vietnam, the study does not explain on what basis it makes the relevant assumptions about the carbon tax and environmental requirements in those two countries. There is also little indication how the numbers are plugged into the above-mentioned free cash flow method to arrive at the enterprise value and internal rate of return. As a result, this not only throws into doubt the accuracy of the quantitative estimates of the model but also raises questions about how useful and realistic these estimates are.
Author Response
Dear reviewer:
We are very grateful to your comments for the manuscript. All of these comments have contributed a lot to improve the quality of our article. According to your advice, we amended the relevant part in manuscript. Some of your questions were answered below.
Point 1: To begin, the paper contains a good number of grammatical errors that make it difficult to follow at times.
Response 1: According to your suggestion, we have corrected the grammatical errors in the revised manuscript.
Point 2: Next, it cites a lot of data without providing sources of the pertinent information. This raises questions about the rigor and accuracy of the study.
Response 2: Thank you for your reminding. We have added data sources for all the data cited in this article( Figure 4, Figure 5, Figure 7,Table 2).
Point 3: Further, coherence is also a problem. While section 3.1 of the study discusses the free cash flow method, the paper fails to show how the method is actually used in establishing the parameters incorporated into the subsequent environmental stress tests.
Response 3: Thank you for your kind comment. In the Appendix B of this paper, we added the calculation process of relevant stress test factors into the operating cost and income.
Point 4: Similarly, the concluding section includes some recommendations that have little to do with the study.
Response 4: We have removed all recommendations that are not relevant to this paper.
Point 5: Moreover, when discussing the GH EMM Power Plant in Indonesia on page 6, the authors veered to a discussion of South Jiangsu region, which is abrupt and perplexing.
Response 5: Thank you for your careful revise. It should be the South Sumatra region here. We have modified this in the revised manuscript.
Point 6: First, while the literature review provides a long list of relevant studies, it fails to show the gap, debate, or inconsistencies of the literature regarding environmental stress testing in China. Thus, it is thus unclear how this study contributes to the literature.
Response 6: This paper introduces the previous literatures from three aspects(stress test method, overseas power investment, power development in Indonesia and Vietnam). Based on your comment, we made the following modifications in the Section 2:
(1) In terms of cited literatures, we reduced literatures related to stress test in the financial sector, added some literatures related to power investment risks, and adjusted the order of some literatures.
(2) We added the relevant summaries of previous literatures in each part. The first part points out the feasibility of the stress test method, the second part points out that the impact of environmental risks on overseas coal power projects is in a blank state, and the third part points out that Indonesia and Vietnam have a huge coal power development market.
(3) We added the relevant contributions of this paper in the last paragraph.
Point 7: Second, and more importantly, while constructing an optimistic and pessimistic scenario when conducting the environmental stress tests on the two select coal-fired power plants in Indonesia and Vietnam, the study does not explain on what basis it makes the relevant assumptions about the carbon tax and environmental requirements in those two countries. There is also little indication how the numbers are plugged into the above-mentioned free cash flow method to arrive at the enterprise value and internal rate of return. As a result, this not only throws into doubt the accuracy of the quantitative estimates of the model but also raises questions about how useful and realistic these estimates are.
Response 7: In the last paragraph of Section 4, we added some relevant reasons for setting carbon tax and environmental tax in Indonesia and Vietnam. And in Appendix B, we added the relevant calculation process.
Thank you again for your advice and hope to learn more from you. If you have any question about this paper, please don’t hesitate to let us know.
Kind regards,
Authors

Round 2
Reviewer 1 Report
The author did not address my following questions.
First, I still do not understand why Vietnam and Indonesia are chosen as the two case studies.
The author replied between 95-99 by stating "China's coal power investment in these two countries is relatively large among "Belt & Road" countries.". please present evidence such as specific figures. you cannot present it by just saying the investment is "relatively large". If you don't have such figure, you may want to consider other significant reasons to assess Vietnam and Indonesia.
Second, by studying only one case in two countries respectively cannot generalise the research outcome. please explain why those two specific projects.
Third, please highlight why your research methodology helps audience understand the situation why China's BRI faces environmental risks. As I pointed out, without your modelling, people can still make the same judgement of the challenges China's companies have been facing. so what is unique about your research methodology?
Fourth, are there any variations between the two cases presented in the paper?
Author Response
Dear reviewer:
We thank you very much for giving us an opportunity to revise our manuscript, and we appreciate you very much for these constructive comments and suggestions on our manuscript. Those comments are all valuable and very helpful for revising and improving our paper, as well as the important guiding significance to our researches. We have studied comments carefully and have made correction which we hope meet with approval.
Revised portion are marked by using the "Track Changes" function in the paper, so that changes are clearly highlighted and easily visible. The main corrections in the paper and the responds to the comments are as follows:
Point 1: First, I still do not understand why Vietnam and Indonesia are chosen as the two case studies.
The author replied between 95-99 by stating "China's coal power investment in these two countries is relatively large among "Belt & Road" countries.". please present evidence such as specific figures. you cannot present it by just saying the investment is "relatively large". If you don't have such figure, you may want to consider other significant reasons to assess Vietnam and Indonesia.
Response 1: Thanks for your comment. Firstly, the main reason why this paper choose Indonesia and Vietnam as case studies is that these two countries are the hot spots for China’s overseas coal power investment in recent years. For this, we have improved the related expression in the Paragraph 2, Section 1. The modified content is as follows.
“South Asia and Southeast Asia are the two primary regions for China’s overseas investment in coal power projects, attributive to their relatively stable political environment, fast-growing economy, and geographical proximity to China. They accounted for 57.11% and 22.75%, respectively, of China's total coal power installations in "Belt & Road" countries. The coal power investment in South Asia was mainly concentrated in India, accounting for 90.35% of the region. But China's participation in India’s coal power is mostly through equipment export, and the investment in coal power projects is mainly concentrated in Southeast Asia. Moreover, due to policy changes in India and economic development in Southeast Asia after 2010, China's participation in coal power generation in South Asia had gradually decreased, and participation in Southeast Asia had continued to increase. By the end of 2016, Indonesia and Vietnam were the first and second largest installed capacity countries in Southeast Asia respectively[5](Appendix A). Chinese finance supporting 30% of all coal-fired capacity under development in Vietnam, and 23% in Indonesia[6]. In recent years, Indonesia and Vietnam have become the hot countries for China to invest coal power projects, which is the main reason for this paper to choose coal power projects in these two countries as the cases.”
In addition,we provided the specific figures about China's overseas coal power investment in Appendix A.
Point 2: Second, by studying only one case in two countries respectively cannot generalise the research outcome. please explain why those two specific projects.
Response 2: Thanks for your comment. Firstly, GH EMM Indonesia power plant and Vietnam Vinhtan power plant are selected in this paper because these two projects have certain representativeness in Indonesia and Vietnam. And we added the following statement at the beginning of Section 4.1.
“This paper selects representative coal-fired power projects in Indonesia and Vietnam for analysis. GH EMM Indonesia Power Plant is the first coal-electricity integration project invested by Shenhua Group overseas[41]. In 2017, it was successfully awarded three awards for Indonesia's “Five Best Power Enterprises”, “Five Best Innovative Power Enterprises” and “Five Best 100MW Power Enterprises”, and won the “Best Innovative Power Enterprise of 2017” for Indonesia. GH EMM Indonesia power plant has shaped a good image of China's power companies. Vietnam Vinhtan coal-fired power plant is the largest coal power project invested by Chinese enterprises in Vietnam[42]. It is also a key production capacity cooperation project in the five-year development plan of China-Vietnam economic and trade cooperation and the five-year plan of onshore infrastructure cooperation[43].”
Secondly, the specific cases and analysis results of this paper are only case studies. What is important is the research framework and general rules highlighted by the cases. And we have emphasized this in the last paragraph of Section 5.
Point 3: Third, please highlight why your research methodology helps audience understand the situation why China's BRI faces environmental risks. As I pointed out, without your modelling, people can still make the same judgement of the challenges China's companies have been facing. so what is unique about your research methodology?
Response 3: Thanks for your comment. Firstly, as we mentioned in the paragraph 2, Section 2. “Most of the existing studies have examined the coal power investment risks in "Belt & Road" countries from the country level perspective of power investment. Some scholars analysed the impact of overseas coal power on the environment. A few scholars studied the impact of climate change on the Chinese overseas coal power projects. However, few works have studied the impact of various environmental risks on the economy of coal power projects invested by Chinese company.”
Based on these studies, this paper comprehensively considered the country risk, power planning risk, climate risk and natural environment risk in the “Belt & Road” countries. In order to quantify the impact of these risks on the economy of coal power projects, this paper select the environmental stress test method. The uniqueness of the stress test method is that it can internalize environmental costs into corporate costs and measure the impact of environmental factors on the value of corporate assets. And we added the corresponding content in the Paragraph 1, Section 3.
Point 4: Fourth, are there any variations between the two cases presented in the paper?
Response 4: Thanks for your comment. The main difference between the two cases lies in the different environmental risks caused by different countries. For this, we added the following contents in the beginning of Section 4.2.
“Due to different countries, GH EMM and Vietnam Vinhtan Power Plant face different environmental risks (Table 1).
(The content of Table 1 can be found in the attachment.)
The following is a detailed analysis of these risks in terms of five stress test factors. ”
In addition, as we mentioned in the last paragraph, Section 5. Although this paper only studies two case projects, the research framework proposed in this paper can provide universal references for Chinese power enterprises.
Thank you again for your advice and hope to learn more from you. If you have any question about this paper, please don’t hesitate to let us know.
Kind regards,
Authors

Round 3
Reviewer 1 Report
I am seriously concerned about the research design. From the start, I do not think it is well designed. As I emphasised previously, by studying only one case in two countries respectively cannot generalise the research outcome. The suggestion is to look at China's coal-fired power plants industry rather than one company or two in the region. It lacks a research question for an academic journal.
second, I am still not convinced why the modelling helps audience reach the conclusion. As I pointed out, without your modelling, people can still make the same judgement of the challenges China's companies have been facing. So what is unique about your research methodology?
Third, maybe a bit of thought-provoking points for the authors. It seems the coal-fired power plants investment in the framework of China's BRI tend to exacerbate the environment in the host country as many media reports have shown. It could be biased but authors need to consider why China is unable to present itself as a responsible player in the sustainable development as it claims to be. from this point of view, China may need to reconsider its push for such investments in the region. the other point of view is that the authors may need to consider the roles of Chinese domestic institutions in Chinese overseas investments. It will contribute to the scholarship if authors could include analysis that part.
in sum, it could be a useful piece for report for consultancy companies or government agencies. however, I do not think it is an academic paper.